



**Do surface lateral flows matter for data assimilation of soil moisture observations**
**into hyperresolution land models?**
**Running title: HYPERRESOLUTION LAND DATA ASSIMILATION**
Yohei Sawada[1,2],
[1] Institute of Engineering Innovation, the University of Tokyo, Tokyo, Japan
[2] Meteorological Research Institute, Japan Meteorological Agency, Tsukuba, Japan
Corresponding author: Y. Sawada, Institute of Engineering Innovation, the University of
Tokyo, Tokyo, Japan, 2-11-6, Yayoi, Bunkyo-ku, Tokyo, Japan, yohei.sawada@sogo.t.u-
tokyo.ac.jp





**Abstract**
It is expected that hyperresolution land modeling substantially innovates the simulation
of terrestrial water, energy, and carbon cycles. The major advantage of hyperresolution
land models against conventional one-dimensional land surface models is that
hyperresolution land models can explicitly simulate lateral water flows. Despite many
efforts on data assimilation of hydrological observations into those hyperresolution land
models, how and when surface water flows driven by local topography matter for data
assimilation of soil moisture observations has not been fully clarified. Here I perform two
minimalist synthetic experiments where soil moisture observations are assimilated into
an integrated surface-groundwater land model by an ensemble Kalman filter. A horizontal
background error covariance provided by overland flows is important to adjust the
unobserved state and parameter variables. However, the non-Gaussianity of the
background error provided by the nonlinearity of a topography-driven surface flow harms
the performance of data assimilation. It is difficult to efficiently constrain model states at
the edge of the area where the topography-driven surface flow reaches by linear-Gaussian
filters, which brings the new challenge in land data assimilation for hyperresolution land
models. This study highlights the importance of surface lateral flows in hydrological data
assimilation.



## 1. Introduction

Hyperresolution land modeling is expected to improve the simulation of terrestrial water,

energy, and carbon cycles, which is crucially important for meteorological, hydrological

and ecological applications (see Wood et al. (2011) for a comprehensive review). While

conventional land surface models (LSMs) assume that lateral water flows are negligible

at the coarse resolution and solve vertical one-dimensional Richards equation for the soil

moisture simulation (e.g., Sellers et al. 1996; Lawrence et al. 2011), currently proposed

hyperresolution land models, which can be applied at a finer resolution (<1km), explicitly

consider surface and subsurface lateral water flows (e.g., Maxwell and Miller 2005; Tian

et al. 2012; Shrestha et al. 2014; Niu et al. 2014). The fine horizontal resolution can

resolve slopes, which are drivers of a lateral transport of water, and realize the fully

integrated surface-groundwater modeling. Previous works indicated that a lateral

transport of water plays an important role in terrestrial water and energy cycles (e.g.,

Maxwell and Condon 2016; Ji et al. 2017; Fang et al. 2017) and land-atmosphere

interactions (e.g., Williams and Maxwell 2011; Keune et al. 2016).



Data assimilation has contributed to improving the performance of LSMs by fusing
simulation and observation. The grand challenge of land data assimilation is to estimate
unobservable variables from observations by propagating observations' information into
model's high dimensional state and parameter space. In previous works on the
conventional 1-D LSMs, many land data assimilation systems (LDASs) have been
proposed to accurately estimate model's state and parameter variables, which cannot be
directly observed, by assimilating satellite and in-situ observations. For example, the
optimization of LSM's unknown parameters (e.g., hydraulic conductivity) has been
implemented by assimilating remotely sensed microwave observations (e.g., Yang et al.
2007; Yang et al. 2009; Bandara et al. 2014; Bandara et al. 2015; Sawada and Koike 2014;
Han et al. 2014). Kumar et al. (2009) focused on the correlation between surface and root-
zone soil moistures to examine the potential of assimilating surface soil moisture
observations to estimate root-zone soil moisture. Sawada et al. (2015) successfully
improved the simulation of root-zone soil moisture by the data assimilation of microwave
brightness temperature observations which include the information of vegetation water
content. Gravity Recovery and Climate Experiment total water storage observation has
been intensively used to improve the simulation of groundwater and soil moisture (e.g.,
Li et al. 2012; Houborg et al. 2012). Improving the simulation of state variables such as



soil moisture and biomass by LDASs has contributed to accurately estimating fluxes such
as evapotranspiration (e.g. Martens et al. 2017) and $CO_2$ flux (e.g., Verbeeck et al. 2011).
However, in most of the studies on the conventional 1-D LDASs, observations impacted
state and parameter variables only in a single model's horizontal grid which is identical
to the location of the observation. The assumption that the water flows are restricted to
vertical direction in LSMs makes it difficult to propagate observation's information
horizontally, which limits the potential of land data assimilation to fully use land
hydrological observations.

The hyperresolution land models, which explicitly solve surface and subsurface lateral
flows, provide a unique opportunity to examine the potential of land data assimilation to
propagate observation's information horizontally in a model space and efficiently use land
hydrological observations. Previous works successfully applied Ensemble Kalman Filters
(EnKF) to 3-D Richards' equation-based integrated surface-groundwater models. For
example, Camporese et al. (2009) and Camporese et al. (2010) successfully assimilated
synthetic observations of surface pressure head and streamflow into the Catchment
Hydrology (CATHY). Ridler et al. (2014) successfully assimilated Soil Moisture and
Ocean Salinity satellite-observed surface soil moisture into the MIKE SHE distributed



hydrological model (see also Zhang et al. (2015)). Kurtz et al. (2016) coupled the Parallel
Data Assimilation Framework (PDAF) (Nerger and Hiller 2013) with the Terrestrial
System Modelling Framework (TerrSysMP) (Shrestha et al. 2014). The performance of
TerrSysMP-PDAF to assimilate soil moisture observations was evaluated by a simple
synthetic experiment (see also Zhang et al. (2018)). Those studies have significantly
contributed to fully assimilating the new high-resolution soil moisture observations such
as Sentinel-1 (e.g., Paroscia et al. 2013)

Although the data assimilation of hydrological observations into hyperresolution land
models has been successfully implemented in the synthetic experiments, it is unclear how
and when topography-driven surface lateral water flows matter for data assimilation of
soil moisture observations. Previous studies on data assimilation with high resolution
models mainly focused on assimilating groundwater observations (e.g., Ait-El-Fquih et
al. 2016; Rasmussen et al. 2015; Hendricks-Franssen et al. 2008). There are some
applications which focused on the observation of soil moisture and pressure head in
shallow unsaturated soil layers. However, in those studies, topography-driven surface
flow has not been considered in the experiment (Kurtz et al. 2016) or the role of them in
assimilating observations into the hyperresolution land models has not been quantitatively





discussed (Camporese et al. 2010; Camporese et al. 2009). This study aims at clarifying
if surface lateral flows matter for data assimilation of soil moisture observations into
hyperresolution land models by a minimalist numerical experiment.


**2. Methods**
2.1. Model
ParFlow is an open source platform which realizes fully integrated surface-groundwater
flow modeling (Kollet and Maxwell 2006; Maxwell et al. 2015). This parallel simulation
platform has been widely used as a core hydrological module in hyperresolution land
models (e.g., Maxwell and Kollet 2008; Maxwell and Condon 2016; Fang et al. 2017;
Kurtz et al. 2016; Maxwell et al. 2011; Williams and Maxwell 2011; Shrestha et al. 2014).
A brief description on the method of ParFlow to simulate integrated surface-subsurface
water flows can be found below and the complete description of ParFlow can be found in
Kollet and Maxwell (2006), Maxwell et al. (2015) and references therein.

In the subsurface, ParFlow solves the variably saturated Richards equation in three
dimensions.





$S_S S_W(h) \frac{\partial h}{\partial t} + \phi S_W(h) \frac{\partial S_W(h)}{\partial t} = \nabla \cdot \mathbf{q} + q_r$    (1)
$\mathbf{q} = -\boldsymbol{K_s}(\boldsymbol{x}) k_r(h) [\nabla(h + z) cos\theta_x + sin\theta_x]$    (2)
In equation (1), $h$ is the pressure head [L]; z is the elevation with the z axis specified as
upward [L]; $S_S$ is the specific storage [L$^{-1}$]; $S_W$ is the relative saturation; $\phi$ is the
porosity [-]; $q_r$ is a source/sink term. Equation (2) describes the flux $\mathbf{q}$
[LT$^{-1}$] by Darcy's law, and $\boldsymbol{K_s}$ is the saturated hydraulic conductivity tensor [LT$^{-1}$]; $k_r$
is the relative permeability [-]; $\theta$ is the local angle of topographic slope (see Maxwell et
al. 2015). In this paper, the saturated hydraulic conductivity is assumed to be isotropic
and a function of z:
$\boldsymbol{K_s} = K_s(z) = K_{s,surface} \exp\left(-f\left(z_{surface} - z\right)\right)$    (3)
where $K_{s,surface}$ is the saturated hydraulic conductivity at the surface soil, and $z_{surface}$
is the elevation of the soil surface. The saturated hydraulic conductivity decreases
exponentially as the soil depth increases (Beven 1982). A van Genuchten relationship
(van Genuchten 1980) is used for the relative saturation and permeability functions.

Overland flow is solved by the two-dimensional kinematic wave equation. The dynamics
of the surface ponding depth, h [L], can be described by:
$\mathbf{k} \cdot [-K_s(z) k_r(h) \cdot \nabla(h + z)] = \frac{\partial \|h, 0\|}{\partial t} - \nabla \cdot \|h, 0\| \boldsymbol{v_{sw}} + q_r$    (4)



In equation (4),  $\mathbf{k}$  is the unit vector in the vertical and  $\|h, 0\|$  indicates the greater value
of the two quantities following the notation of Maxwell et al. (2015). If h < 0, equation
(4) describes that vertical fluxes across the land surface is equal to the source/sink term
$q_r$  (i.e., rainfall and evapotranspiration). If h > 0, the terms on the right-hand side of
equation (4), which indicate water fluxes routed according to surface topography, are
active.  $\boldsymbol{v_{sw}}$  is the two-dimensional depth-averaged water flow velocity [LT$^{-1}$] and
estimated by the Manning's law:
$$\boldsymbol{v_{sw}} = \begin{pmatrix} \frac{\sqrt{S_{f,x}}}{n} h^{\frac{2}{3}} \\ \frac{\sqrt{S_{f,y}}}{n} h^{\frac{2}{3}} \end{pmatrix} \ (5)$$
where  $S_{f,x}$  and  $S_{f,y}$  are the friction slopes [-] for the x- and y-direction, respectively; n
is the Manning's coefficient [TL$^{-1/3}$]. In the kinematic wave approximation, the friction
slopes are set to the bed slopes. The methodology of discretization and numerical method
to solve equations (1-5) can be found in Kollet and Maxwell (2006).


**2.2. Data Assimilation**
In this paper, the ensemble Kalman filter (EnKF) was applied to assimilate soil moisture
observations into ParFlow. The general description of the Kalman filter is the following:
$\boldsymbol{x}^f(t) = \mathcal{M}[\boldsymbol{x}^a(t-1)]$  (6)





$x^a(t) = x^f(t) + K[y^o - \mathcal{H}x^f(t)]$ (7)
$K = P^f \mathcal{H}^T (\mathcal{H} P^f \mathcal{H}^T + R)^{-1}$ (8)
$P^a = (I - K\mathcal{H})P^f$ (9)
I follow the notation of Houtekamer and Zhang (2016). In equation (6), a forecast model
$\mathcal{M}$ (ParFlow in this study) is used to obtain a prior estimate at time t, $x^f(t)$, from the
estimation at the previous time $x^a(t-1)$. In equation (7), a prior estimate $x^f(t)$ is
updated to the analysis state, $x^a(t)$, using new observations $y^o$. The Kalman gain matrix
**K,** calculated by equation (8), gives an appropriate weight for the observations with an
error covariance matrix $R$, and the prior with an error covariance matrix $P^f$. To calculate
**K**, the observation operator $\mathcal{H}$ is needed to map from model space to observation space.
It should be noted that the equations (6-9) give an optimal estimation only when the model
and observation errors follow the Gaussian distribution. When the probabilistic
distribution of the error in either model or observation has a non-Gaussian structure,
results of the Kalman filter are suboptimal. This point is important to interpret the results
of this study.

EnKF is the Monte Carlo implementation of equations (6-9). To compute the Kalman gain
matrix, **K,** ensemble approximations of $P^f \mathcal{H}^T$ and $\mathcal{H} P^f \mathcal{H}^T$ can be given by:



$\boldsymbol{P^f \mathcal{H}^T} \equiv \frac{1}{k-1}\sum_{i=1}^{k}\left(\boldsymbol{x_i^f} - \overline{\boldsymbol{x^f}}\right)\left(\mathcal{H}\boldsymbol{x_i^f} - \overline{\mathcal{H}\boldsymbol{x^f}}\right)^T$ (10)
$\boldsymbol{\mathcal{H}P^f \mathcal{H}^T} \equiv \frac{1}{k-1}\sum_{i=1}^{k}\left(\mathcal{H}\boldsymbol{x_i^f} - \overline{\mathcal{H}\boldsymbol{x^f}}\right)\left(\mathcal{H}\boldsymbol{x_i^f} - \overline{\mathcal{H}\boldsymbol{x^f}}\right)^T$ (11)
where $\boldsymbol{x_i^f}$ is the ith member of a k-member ensemble prior and $\overline{\boldsymbol{x^f}} = \frac{1}{k}\sum_{i=1}^{k}\boldsymbol{x_i^f}$ and
$\overline{\mathcal{H}\boldsymbol{x^f}} = \frac{1}{k}\sum_{i=1}^{k}\mathcal{H}\boldsymbol{x_i^f}$.

Once $\overline{\boldsymbol{x^a}} = \sum_{i=1}^{k}\boldsymbol{x_i^a}$ ($\boldsymbol{x_i^a}$ is the ith member of a k-member ensemble analysis) and $\boldsymbol{P^a} =$
$\frac{1}{k-1}\sum_{i=1}^{k}(\boldsymbol{x_i^a} - \overline{\boldsymbol{x^a}})(\boldsymbol{x_i^a} - \overline{\boldsymbol{x^a}})^T$ are computed by equations (6-11), there are many
choices of an analysis ensemble. Although equations (6-11) can calculate the mean and
variance of the ensemble members, they do not tell how to adjust the state of the ensemble
members in order to realize the estimated mean and variance. There are many proposed
flavors of EnKF and one of the differences among them is the method to choose the
analysis $\boldsymbol{x_i^a}$. In this paper, the Ensemble Transform Kalman Filter (ETKF; Bishop et al.
2001; Hunt et al. 2007) was used to transport forecast ensembles to analysis ensembles.
Please refer to Hunt et al. (2007) for the complete description of the ETKF and its
localized version, the Local Ensemble Transform Kalman Filter (LETKF). The open
source available at https://github.com/takemasa-miyoshi/letkf was used in this study as
the ETKF code library.



In many ensemble Kalman filter systems, the ensemble spread tends to become
underdispersive without any ensemble inflation methods (Houtekamer and Zhang, 2016).
In this paper, the relaxation to prior perturbation method (RTPP) of Zhang et al. (2004)
was used to maintain an appropriate ensemble spread. In the RTPP, the computed analysis
perturbations are relaxed back to the forecast perturbations:
$\boldsymbol{x}_{i,new}^{a} = (1-\alpha)(\boldsymbol{x}_i^a - \overline{\boldsymbol{x}^a}) + \alpha(\boldsymbol{x}_i^f - \overline{\boldsymbol{x}^f}),\ 0 \leq \alpha \leq 1$  (12)
where $\alpha$  was set to 0.975 in this study.

In the data assimilation experiments, I adjusted pressure head by data assimilation so that
$\boldsymbol{x}^f$  is pressure head. Since the surface saturated hydraulic conductivity was also adjusted,
$\boldsymbol{x}^f$  includes log-transformed $K_{s,surface}$. Since I assimilated volumetric soil moisture
observations ($\boldsymbol{y}^o$  are observed volumetric soil moisture), the van Genuchten relationship
works as an observation operator $\mathcal{H}$  in order to transport the model estimated pressure
head into the observable volumetric soil moisture in this study.


**2.3. Kullback-Leibler divergence**

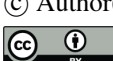



To evaluate the non-Gaussianity of the background error sampled by an ensemble, I used
the Kullback-Leibler divergence (KLD) (Kullback and Leibler 1951):
$D_{KL}(p,q) = \sum_i p(i) log \frac{p(i)}{q(i)}$ (13)
where $D_{KL}(p,q)$ is the KLD between two probabilistic distribution functions (PDFs), $p$
and $q$. If two PDFs are equal for all $i$, $D_{KL}(p,q) = 0$. A large value for $D_{KL}(p,q)$
indicates that the two PDFs, $p$ and $q$, substantially differ from each other. Therefore,
the KLD can be used   as an index to evaluate the closeness of two PDFs. It should be
noted that the KLD is not symmetric ($D_{KL}(p,q) \neq D_{KL}(q,p)$).


**3. Synthetic experiments**
**3.1. Simple 2-D slope with homogeneous hydraulic conductivity**
**3.1.1. Experiment Design**
The synthetic experiment was implemented to examine how topography-driven surface
lateral flows contribute to efficiently propagating observation's information horizontally
in the data assimilation of soil moisture observation. Two synthetic reference runs were
created by Parflow. The 2-D domain has a horizontal extension of 4000m and a vertical
extension of 5m. The domain of the virtual slope was horizontally discretized into 40 grid



cells with a size of 100m and vertically discretized into 50 grid cells with a size of 0.10m.
The domain has a 25% slope. In two synthetic reference runs, it heavily rains only in the
upper half of the slope (2000m<x<4000m). A constant rainfall rate of 50mm/h was
applied for 3 hours and then the period with no rainfall and evaporation of 0.075mm/h
lasted for 117 hours. This 120-hour rain/no rain cycle was repeatedly applied to the
domain. There is no rainfall in the lower half of the slope (0m<x<2000m). The
configurations described above were schematically shown in Figure 1a. The parameters
of the van Genuchten relationship, alpha and n, were set to 1.5 [m$^{-1}$] and 1.75, respectively.
The porosity, $\phi$ in equation (1), was set to 0.40. The Manning's coefficient, n in equation
(5), was set to $5.52 \times 10^{-6}$ [m$^{-1/3}$h]. These clayey soil properties described above are
applied to the whole domain. The groundwater table was located at z=3m and the
hydrostatic pressure gradient was assumed for the initial pressure heads in the unsaturated
soil layers.

The difference between two synthetic reference runs is the value of saturated hydraulic
conductivity. The surface saturated hydraulic conductivity, $K_{s,surface}$ in equation (3),
was set to 0.005 [m/h] in one reference, and 0.02 [m/h] in the other. These surface
saturated hydraulic conductivities described above are applied to the whole domain.


Figure 1 shows the difference of the response to heavy rainfall between the two synthetic
reference runs. In the case of the low saturated hydraulic conductivity (hereafter called
the LOW_K reference), larger surface lateral flows are generated than the case of the high
saturated hydraulic conductivity (hereafter called the HIGH_K reference). In the LOW_K
reference, the topography-driven surface lateral flows reach the left edge of the domain
(Figure 1b). In the HIGH_K reference, supplied water moves vertically rather than
horizontally and the topography-driven surface flow reaches around x = 1000~1500m
(Figure 1d).

For the data assimilation experiment, an ensemble of 50 realizations was generated. Each
ensemble member has different saturated hydraulic conductivity and rainfall rate.
Lognormal multiplicative noise was added to surface saturated hydraulic conductivity
and rainfall rate of the synthetic reference runs. This specification of uncertainty in
rainfall was also adopted in Crow et al. (2011). The two parameters of the lognormal
distribution, commonly called $\mu$ and $\sigma$, were set to 0 and 0.15, respectively. The initial
groundwater depth of each ensemble member was drawn from the uniform distribution
from 2.0m to 3.5m. The hydrostatic pressure gradient was assumed for the initial pressure
heads in the unsaturated soil layers.




The virtual hourly observations were generated by adding the Gaussian white noise whose
mean is zero to the volumetric soil moisture simulated by the synthetic reference runs.
The observation error (the standard deviation of the added Gaussian white noise) was set
to 0.05 m³/m³. It was assumed that the volumetric soil moistures can be observed in every
model's soil layer from surface to the depth of 1m at the specific location. The two
scenarios of the observation's location are provided. In the first scenario (hereafter called
the UP_O scenario), the volumetric soil moisture at the upper part of the slope (x =
2500m) was observed. In the UP_O scenario, I could observe the volumetric soil moisture
in the upper part of the slope where it heavily rains and tried to infer the soil moisture in
the lower part of the slope where it does not rain by propagating the observation's
information downhill. In the second scenario (hereafter called the DOWN_O scenario),
the volumetric soil moisture at the lower part of the slope (x = 1500m) was observed. In
the DOWN_O scenario, I could observe the volumetric soil moisture in the lower part of
the slope where it does not rain and tried to infer the soil moisture in the upper part of the
slope where it heavily rains by propagating the observation's information uphill.





Since I had the two synthetic reference runs (the HIGH_K and LOW_K references) and
the two observation scenarios (the UP_O and DOWN_O scenarios), I implemented totally
four data assimilation experiments. Table 1 summarizes the data assimilation experiments
implemented in this study. For instance, in the HIGH_K-UP_O experiment, I chose the
HIGH_K reference and generated an ensemble of 50 realizations from the HIGH_K
reference. The soil moisture observations were generated from the HIGH_K reference at
the location of x = 2500m and assimilated into the model every hour. The simulated
volumetric soil moisture of the data assimilation experiment was compared with that of
the HIGH_K reference.

In addition to the data assimilation (DA) experiments, I implemented the NoDA
experiment (also called the open-loop experiment in the literature of the LDAS study) in
which the ensemble was used but no observation data were assimilated. As evaluation
metrics, root-mean-square-error (RMSE) was used:
$\text{RMSE} = \sqrt{\frac{1}{k}\sum_{i=1}^{k}(F_i - T)^2}$  (14)
where k is the ensemble number, $F_i$ is the volumetric soil moisture simulated by the i-th
member in the DA or NoDA experiment, T is the volumetric soil moisture simulated by
the synthetic reference run.




To evaluate the impact of data assimilation, the improvement rate (IR) was defined and
calculated by the following equation:
$\text{IR} = \dfrac{\overline{RMSE_{DA}} - \overline{RMSE_{NoDA}}}{\overline{RMSE_{NoDA}}}$ (15)
where $\overline{RMSE_{DA}}$ and $\overline{RMSE_{NoDA}}$ are time-mean RMSE of the DA and NoDA
experiments, respectively. The negative IR indicates that data assimilation positively
impacts the simulation of soil moisture. The metrics described above was calculated in
the whole domain. In the DA experiment, soil moisture values before the update by ETKF
(i.e. initial guess) were used to calculate the metrics.

Four of 120-hour rain/no rain cycles were applied so that the computation period was 480
hours. The spin-up results in the first 120 hours were not used to calculate the evaluation
metrics. Since the steady state of groundwater level is not the scope of this paper, the long
spin-up is not absolutely necessary.


**3.1.2. Results**





Figure 2a shows the IR of the LOW_K-UP_O experiment. The time series of the DA and
NoDA experiment and the synthetic reference run in the LOW_K-UP_O experiment can
be found in Figure S1. The data assimilation efficiently propagates the information of the
observations located in the upper part of the slope (see the black arrow in Figure 2a) both
horizontally and vertically. RMSE is reduced by data assimilation not only directly under
the observation but also the lower part of the slope where it does not rain. The optimized
$K_{s,surface} \approx 0.00508$ [m/h]  is also accurate. However, the increase of RMSE by data
assimilation can be found at the left edge of the domain, which is far from the location of
the observation. The impact of data assimilation on the surface soil moisture simulation
is small because the RMSE of the NoDA experiment is already small ($\leq 0.01 \mathrm{m}^3/\mathrm{m}^3$) there
in the case of the LOW_K reference so that any improvements there do not make sense.

Figure 2b shows the IR of the LOW_K-DOWN_O experiment (see also Figure S2 for
time series). The IR's spatial pattern of the LOW_K-DOWN_O experiment is similar to
that of the LOW_K-UP_O experiment. It is promising that I can accurately infer soil
moisture in the region where it heavily rains from the shallow soil moisture observations
in the region where it does not rain. The optimized $K_{s,surface} \approx 0.00512$ [m/h] is also
accurate.






Figure 3a shows the difference of time-mean RMSEs ($\overline{RMSE_{DA}}$ in equation (15))
between the LOW_K-UP_O and LOW_K-DOWN_O experiments. Although observing
the lower part of the slope slightly improves the soil moisture simulation at the left edge
of the domain compared with observing the upper part of the slope, there are few
differences between the UP_O and DOWN_O scenarios in the case of the LOW_K
reference. The soil moisture observations have large representativeness and I can
efficiently infer soil moisture in the soil columns which are horizontally and vertically far
from the observations.

Figure 2c shows the IR of the HIGH_K-UP_O experiment (see also Figure S3 for time
series). The data assimilation significantly reduces RMSE of the soil moisture simulation
directly under the observations (see the black arrow in Figure 2c), which indicates that
the data assimilation efficiently propagates the information of the observations vertically.
The saturated hydraulic conductivity is also accurately optimized ($K_{s,surface} \approx 0.0204$
[m/h]. However, the impact of the data assimilation on the soil moisture simulation in the
lower part of the slope around x=1500m is marginal although there are large RMSE in





the NoDA experiment (>0.05m³/m³) at the edge of the area where topography-driven
surface flow reaches in the HIGH_K reference (see Figure 1d).

Figure 2d shows the IR of the HIGH_K-DOWN_O experiment (see also Figure S4 for
time series). Although the observations in the lower part of the slope (see the black arrow
in Figure 2d) significantly improve the soil moisture simulation in the downstream area
of the observation and accurately optimize $K_{s,surface} \approx 0.0208$ [m/h], the impact of the
data assimilation on the shallow soil moisture simulation around x=500~1000m is
marginal. As I found in the LOW_K-DOWN_O experiment, the shallow soil moisture
observations in the region where it does not rain can improve the soil moisture simulation
in the region where it heavily rains. However, the IR of the HIGH_K-DOWN_O
experiment in the upper part of the slope is smaller than that of the LOW_K-DOWN_O
experiment (see Figure 2b and 2d).

The high representativeness of the observations which I found in the case of the LOW_K
reference cannot be found in the case of the HIGH_K reference. Figure 3b shows the
difference of time-mean RMSEs ($\overline{RMSE_{DA}}$ in equation (15)) between the HIGH_K-
UP_O and HIGH_K-DOWN_O experiments. Compared with the LOW_K reference case





(Figure 3a), there are significant differences between the UP_O and DOWN_O scenarios
in the case of higher saturated hydraulic conductivity. In this case, the vertical propagation
of the observations' information is more efficient than the horizontal propagation.

The relatively low efficiency of the data assimilation and the low representativeness of
the soil moisture observations in the case of the HIGH_K reference are caused by the
non-Gaussian background error distribution. I calculated KLD by comparing the PDF of
the NoDA ensemble ($p$ in equation (13)) with the Gaussian PDF which has the mean and
variance of the NoDA ensemble ($q$ in equation (13)). Figure 4 shows that the NoDA
ensemble in the case of the HIGH_K reference has stronger non-Gaussianity than the case
of the LOW_K reference especially in the shallow soil layers. The strong non-Gaussianity
of the NoDA ensemble generated from the HIGH_K reference can be found at the edge
of the area where the topography-driven surface flow reaches (Figure 1d). Figure 5 shows
that there is the bifurcation of the ensemble in this region when the ensemble is generated
from the HIGH_K reference. The process of topography-driven surface flows is switched
on if and only if the surface soil is saturated (see equation (4)) so that the ensemble tends
to be bifurcated into the members with surface flows and without surface flows. As I
mentioned in section 2.2, in the ETKF, the state and parameter variables are adjusted



assuming the Gaussian PDF of the model's error and the linear relationship between
observed variables and unobserved variables. Therefore, the non-Gaussianity of the prior
ensemble induced by the strong non-linear dynamics of surface lateral flows makes the
ETKF inefficient. It is more difficult to reconstruct 3-D fields of soil moisture in high
conductivity soils since the 1-D vertical water movement is more dominant. The absolute
RMSE of the NoDA experiment in the HIGH_K reference is larger than the LOW_K
reference in many places (not shown). Please note that the non-Gaussianity can also be
found in the LOW_K reference at the edge of the domain (x=500m) due to the non-linear
dynamics, which causes the degradation of the soil moisture simulation in the LOW_K-
UP_O experiment (see Figure 2a).

One of the major simplifications in this experiment is spatially homogeneous surface
saturated hydraulic conductivity. The optimization of it can efficiently improve the soil
moisture simulation in the whole domain. However, the optimization of this
homogeneous surface saturated hydraulic conductivity has a limited impact on the soil
moisture simulation. Figure S5 shows the IR of the HIGH_K-DOWN_O experiment
where the parameter optimization by ETKF is switched off. Even if I do not optimize the
surface saturated hydraulic conductivity, I could obtain the similar IR to the original



experiment and the shallow soil moisture observations in the region where it does not rain
can improve the soil moisture simulation in the region where it heavily rains. The
horizontal propagation of the observations' information shown in this experiment was
brought out not only by the optimization of spatially homogeneous saturated hydraulic
conductivity but also by the horizontal error correlation due to topography-driven surface
flows.

Please note that the improvement of the soil moisture simulation cannot be found if the
topography-driven surface flow is neglected. Figure S6 shows the IR of the LOW-
K_DOWN-O experiment where the topography-driven surface flow is neglected in the
ParFlow simulation. The imperfect model physics of ParFlow substantially degrades the
skill to simulate soil moisture and data assimilation cannot compensate this degradation.
This point will also be discussed in the section 3.2 more deeply.

**3.2. Simple 3-D slope with heterogeneous hydraulic conductivity**
**3.2.1. Experiment design**
To further demonstrate how land data assimilation works with topography-driven surface
lateral flows, I implemented another synthetic experiment which is more realistic than



that shown in section 3.1. The 3-D domain has a horizontal extension of 4000 m×4000m
and a vertical extension of 3m. The domain was horizontally discretized into 40×40 grid
cells with a size of 100m×100m and vertically discretized into 30 grid cells with a size
of 0.1m. The domain has a 10% slope in both x and y directions (see Figure 6a). The
parameters of the van Genuchten relationship, porosity and Manning's coefficient were
set to the same variables for the synthetic experiment in section 3.1.

The spatially heterogeneous surface saturated hydraulic conductivity was generated
following Kurtz et al. (2016). The field of $log_{10}(K_{s,surface})$ was generated by two-
dimensional unconditioned sequential Gaussian simulation. A Gaussian variogram with
nugget, sill, and range values of  0.0 $log_{10}$(m/h), 0.1 $log_{10}(m^2h^2)$, and 12 model
grids (1200m), respectively was used to simulate the spatial distribution of
$log_{10}(K_{s,surface})$. A constant value of -2.30 $log_{10}$(m/h) (i.e. 0.005 (m/h)) was added
to the generated field. Subsurface saturated hydraulic conductivity was calculated by
equation (3). An ensemble of 51 realizations of $log_{10}(K_{s,surface})$ was generated and one
of them was chosen as a synthetic reference (Figure 6a). The remaining 50 members were
used for data assimilation experiments.



A rainfall rate $R(x, y)$ (mm/h) was modelled by a logistic function:
$R(x, y) = \frac{R_{max}}{1+100\exp\left(-0.2 \times \frac{x+y}{2}\right)}$ (16)
where x and y are horizontal grid numbers ($1 \leq x \leq 40, 1 \leq y \leq 40$). In the synthetic
reference, the maximum rainfall rate in the domain, $R_{max}$, was set to 50 (mm/h) (Figure
6b). This rainfall rate was applied for 3 hours and then the period with no rainfall and
evaporation of 0.075mm/h lasted for 117 hours. For data assimilation experiment, an
ensemble of 50 realization of $R(x, y)$ was generated by adding a lognormal
multiplicative noise to $R_{max}$ of the synthetic reference. The two parameters of the
lognormal distribution, commonly called $\mu$ and $\sigma$, were set to 0 and 0.15, respectively.

Figure 6c shows the distribution of surface soil moisture in the synthetic reference run.
Strong rainfall rate applied in the upper part of the slope generates the topography-driven
surface lateral flows. The virtual hourly observations were generated by adding the
Gaussian white noise, whose mean is zero and standard deviation is 0.05 m$^3$/m$^3$, to the
volumetric surface soil moisture simulated by the synthetic reference run. Unlike the
experiment in section 3.1, only surface soil moisture can be observed in this synthetic
experiment, which makes this experiment more realistic since satellite sensors can
observe only surface soil moisture. Three different observing networks with different



observation densities were used (Figure 7). The observing networks shown in Figure 7a,
7b, and 7c have totally 1, 9, and 361 observations and are called obs1, obs9, and obs361,
respectively.

In the DA experiments, those virtual observations of surface soil moisture were
assimilated every hour to adjust pressure head and saturated hydraulic conductivity. As I
did in the section 3.1, the NoDA experiments were also implemented. The two different
configurations of ParFlow were used for both DA and NoDA experiments. In the first
configuration, called OF, Parflow explicitly solves overland flows. In the second
configuration, called noOF, Parflow assumes the flat terrain for surface flows so that no
overland flows are generated. Since the synthetic reference run explicitly considers the
topography-driven surface flow, the configuration of noOF assumes that the model
physics is imperfect. I implemented 8 numerical experiments which are summarized in
Table 2. For example, the OF_DA_obs9 experiment is the data assimilation experiment
with the observing network shown in Figure 7b, in which Parflow explicitly solves the
topography-driven surface flow. The noOF_NoDA is the model run without assimilating
observations, in which Parflow does not consider the topography-driven surface flow.






### 3.2.2. Results

Figure 8a shows the RMSE of soil moisture simulation of a second soil layer (i.e. 10-
20cm soil depth) in all 8 experiments (the same conclusion described below can be
obtained by analyzing all of shallow soil layers). When Parflow explicitly solves the
topography-driven surface flow, data assimilation substantially reduces RMSE of the soil
moisture simulation (green bars in Figure 8a). The OF_DA_obs361 experiment has the
smallest RMSE so that a denser observing network is beneficial to estimate soil moisture.
Figure 8b shows the RMSE of the estimation of saturated surface hydraulic conductivity
in all 8 experiments. Data assimilation also reduces the uncertainty in model's parameters
(green bars in Figure 8b). However, the OF_DA_obs361 experiment has larger RMSE
than the other DA experiments. This is because the adjustment of hydraulic conductivity
in the OF_DA_obs361 experiment is overfitting to observations. In the OF configuration,
there are two sources of errors, rainfall rate and hydraulic conductivity. However, data
assimilation can adjust only hydraulic conductivity so that the assimilation of a large
number of observations causes overfitting to mitigate the impact of errors in rainfall rate.



The noOF_NoDA experiment has larger RMSE than the OF_NoDA experiment due to
the negligence of the topography-driven surface flow. In the noOF configuration, data
assimilation also improves the soil moisture simulation (red bars in Figure 8a). The
noOF_DA_obs361 experiment outperforms the OF_NoDA experiment so that data
assimilation with a dense observing network can compensate the negative impact of
neglecting the topography-driven surface flow. Although data assimilation positively
impacts the parameter estimation, the denser observing network cannot reduce RMSE of
hydraulic conductivity estimation (red bars in Figure 8b). The negative impact of the
dense observations in the noOF_DA_obs361 experiment on the parameter estimation is
larger than in the OF_DA_obs361 experiment. In addition to rainfall rate and hydraulic
conductivity, the imperfect model physics (i.e., no topography-driven surface flow) is the
source of error in the noOF configuration. The assimilation of a large number of
observations causes overfitting because it mitigates the impact of all systematic errors
which comes from three different sources only by adjusting hydraulic conductivity.

Figure 9 shows the difference of RMSE of the soil moisture simulation between the DA
experiments and the OF_NoDA experiment. In the DA configuration, the improvement
of the soil moisture estimation can be found in a large area even if there is a single





observation in the center of the domain (Figure 9a). Figure 9b shows that the increase of
the number of observations substantially improves the soil moisture simulation in the
region which is affected by topography-driven surface flow (see also Figure 6c). However,
the skill to simulate soil moisture is severely degraded in the lower-left corner of the
domain, which causes the stalled improvement from the OF_DA_obs1 experiment to the
OF_DA_obs9 experiment shown in Figure 8a. Figure 9c shows that although the far
denser observing network can slightly mitigate this degradation, increasing the number
of observations cannot efficiently solve this issue. This degradation is caused by the
bifurcation of ensemble members at the edge of the area where topography-driven surface
flow reaches (Figure S7). Figure 10 shows KLD in the OF_NoDA and noOF_NoDA
experiments. Figure 10a clearly shows that the ensemble simulation generates the strong
non-Gaussianity at the edge of the area where topography-driven surface flow reaches,
which harms the efficiency of the ETKF. This finding is consistent to what I found in the
previous experiment in section 3.1.

In the noOF configuration, there are large errors in the area around 500<=x, y <=1500
since the increase of soil moisture in this area is caused by the topography-driven surface
flow which is neglected in the noOF configuration. Figures 9d and 9e show that the sparse





observations cannot completely remove this degradation caused by imperfect model
physics. Figure 9f shows that the noOF_DA_obs361 can outperform the OF_NoDA
experiment in exchange for the degradation of the parameter estimation as I found in
Figure 8. The unstable behavior of the ETKF found in the OF configuration does not
occur when the topography-driven surface flow is neglected since the ensemble
simulation does not generate the non-Gaussian background distribution (Figure 10b).



**4. Discussion**
In this study, I revealed that the hyperresolution integrated surface-subsurface
hydrological model gives the unique opportunity to effectively use soil moisture
observations to improve the soil moisture simulation in terms of a horizontal propagation
of observation's information in a model space. I found that the explicit calculation of the
topography-driven surface flow has an important role in propagating the information of
soil moisture observation horizontally by data assimilation even if there is considerable
heterogeneity of meteorological forcing. It is possible that the soil moisture observations



in the area where it does not heavily rain can improve the soil moisture simulation in the
severe rainfall area.

This potential cannot be brought out in the conventional 1-D LSM where sub-grid scale
surface runoff is parameterized and the surface flows in one grid do not move to the
adjacent grids. Neglecting the topography-driven surface flow causes significant bias in
the soil moisture simulation and this bias cannot be completely mitigated by data
assimilation especially in the case of a sparse observing network. However, I found that
even if the model uses imperfect physics which neglects the interaction between
topography-driven surface lateral flows and subsurface soil moisture, assimilating soil
moisture observations into the model's three-dimensional state and parameter space can
improve the skill to estimate soil moisture and hydraulic conductivity. This finding
implies that the conventional 1-D LSM with full 3-D data assimilation may be a
computationally cheap and reasonable choice in some cases although many land data
assimilation systems with the conventional 1-D LSM currently update state variables only
in a single model's horizontal grid which is identical to the location of the observation.





The conventional ensemble data assimilation (i.e. ETKF) severely suffers from the non-
Gaussian background error PDFs caused by the strongly nonlinear dynamics of the
topography-driven surface flow. The efficiency of ETKF to propagate the information of
observations horizontally in the model space is limited in the edge of the area where the
topography-driven surface flow reaches. Please note that the low representativeness of
the soil moisture observations in the case of the HIGH_K reference shown in section 3.1
is due to the core assumption of the Kalman filter that the error PDFs follow the Gaussian
distribution so that the increase of the ensemble size cannot solve this issue. I
implemented the data assimilation experiment in the case of the HIGH_K reference with
an ensemble size of 500, which is 10 times larger than the original experiments shown in
section 3.1, and found no significant improvement of the soil moisture simulation (not
shown). Some studies revealed that volumetric soil moisture distributions follow the
Gaussian distribution better than pressure head so that they recommend to update soil
moisture as a state variable (e.g., Zhang et al. (2018)). However, in this study, I found that
volumetric soil moisture distributions have bimodal structure and do not follow the
Gaussian distribution. The limitation of ensemble Kalman filters found in this study does
not depend on the updated state variables.



The spatially dense soil moisture observations are needed to efficiently constrain state
variables at the edge of surface flows. High resolution soil moisture remote sensing based
on satellite active and passive combined microwave observations at the 1 km spatial
resolution (e.g., He et al. 2018) and the assimilation of those data (Lievens et al. 2017)
may be important in the era of the hyperresolution land modeling.   High resolution
observations of surface inundated water from satellite imagery with a spatial resolution
finer than 100 m (e.g., Sakamoto et al. 2007; Arnesen et al. 2013) may also be useful.
However, the numerical experiment in section 3.2 implies that the dense observing
network of surface soil moisture cannot completely remove the negative impact of the
non-Gaussian background PDF.

Since there is a nonlinear relationship between observed and unobserved variables
sampled by an ensemble, a localization method, which spatially restricts the impact of
assimilating observations, is crucially needed for real-world applications. The results of
this study imply that the optimal localization radius strongly depends on the model
parameter (i.e. saturated hydraulic conductivity). Rasmussen et al. (2015) successfully
applied the adaptive localization method (Anderson 2007; Bishop and Hodyss 2009) to
the data assimilation of groundwater observations into a hydrological model. It is

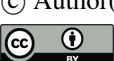



appropriate to adaptively determine the localization radius considering the lack of prior
knowledge of how soil moisture simulated by an ensemble is horizontally correlated.

Reducing the uncertainty in rainfall positively impacts the efficiency of data assimilation
since the bifurcation of simulated soil moisture found in Figure 5c is originally induced
by the uncertainty in rainfall. Although assimilating land hydrological observations to
improve the rainfall input has been intensively investigated (e.g., Sawada et al. 2018;
Herrnegger et al. 2015; Crow et al. 2011; Vrugt et al. 2008), it has yet to be applied to
hyperresolution land models. Please note that the parameters of the lognormal distribution
to model the uncertainty in rainfall were specified to make the rainfall PDF similar to the
Gaussian distribution. I chose the lognormal distribution in order not to generate negative
rainfall values and I intended not to introduce non-Gaussianity into the external forcing.
The rainfall input which follows the Gaussian PDF was transformed into the non-
Gaussian PDF of the background error by the strongly nonlinear dynamics of the
topography-driven surface flow.

To explicitly consider non-Gaussianity and non-linear relationship between observed and
unobserved variables induced by the topography-driven surface flow, the particle filters



may be useful. The particle filter can represent a probability distribution (including non-
Gaussian distributions) directly by an ensemble. Particle filters have been intensively
applied to conventional 1-D LSMs (e.g., Sawada et al. 2015; Qin et al. 2009) and lumped
hydrological models (e.g., Yan and Moradkhani 2016; Vrugt et al. 2013). Although
particle filtering in a high dimensional system suffers from the "curse of dimensionality"
(e.g., Snyder et al. 2008), the applicability of particle filtering to 3-D hyperresolution land
models should be assessed in the future.

Since the synthetic numerical experiments in this paper adopted the simple and
minimalistic setting, the findings of this paper may be exaggerated. There are no river
channels in the synthetic experiment so that the skill to simulate river water level and
discharge cannot be discussed, which is the major limitation of this study. The simple
representation of soil properties is also a limitation of this study. In future work, the
contributions of the topography-driven surface runoff process to the data assimilation of
hydrological observations should be quantified in real-world applications. In addition, in
the virtual experiment of this paper, I neglected some of the important land processes such
as transpiration, canopy interception, snow, and frozen soil. Although they are generally
not primary factors in the propagation of overland flows generated by extreme rainfall,



which has a shorter timescale than the neglected processes, those processes should be
considered in the future.


**5. Conclusions**
The simplified synthetic experiments of this study indicate that topography-driven lateral
surface flows induced by heavy rainfalls do matter for data assimilation of hydrological
observations into hyperresolution land models. Even if there is extreme heterogeneity of
rainfall, the information of soil moisture observations can be propagated horizontally in
the model space and the soil moisture simulation can be improved by the ensemble
Kalman filter. However, the nonlinear dynamics of the topography-driven surface flow
induces the non-Gaussianity of the model error , which harms the efficiency of data
assimilation of soil moisture observations. It is difficult to efficiently constrain model
states at the edge of the area where the topography-driven surface flow reaches by linear-
Gaussian filters, which brings the new challenge in land data assimilation for
hyperresolution land models.

**Acknowledgement**





654 This study was supported by the JSPS KAKENHI grant JP17K18352 and 18H03800.


**Code/Data Availability**

657 All data used in this paper are stored in the repository of the University of Tokyo for 5

658 years and available upon request to the author. The ETKF code used in this study can be

659 found at https://github.com/takemasa-miyoshi/letkf.


**Author Contribution**

662 YS designed the study, executed numerical experiments, analyzed the results, and wrote

663 the paper.


**Competing interests**

666 The author declares no competing interests.

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

Availability on Convective-Scale Data Assimilation with an Ensemble Kalman Filter.
*Monthly.    Weather.    Review.,*    **132**,    1238–1253,    https://doi.org/10.1175/1520-
0493(2004)132<1238:IOIEAO>2.0.CO;2, 2004

Zhang, H., Kurtz, W., Kollet, S., Vereecken, H., and Franssen, H. J. H.: Comparison of
different assimilation methodologies of groundwater levels to improve predictions of root
zone soil moisture with an integrated terrestrial system model. *Advances in Water*
*Resources*, *111*, 224–238. https://doi.org/10.1016/j.advwatres.2017.11.003, 2018.









**Table 1.** Configuration of the data assimilation experiments in section 3.1.

|  | hydraulic conductivity [m/h] | observation's location [m] |
|---|---|---|
| LOW_K-UP_O | 0.005 | 2500 |
| LOW_K-DOWN_O | 0.005 | 1500 |
| HIGH_K-UP_O | 0.02 | 2500 |
| HIGH_K-DOWN_O | 0.02 | 1500 |


**Table 2.** Configuration of the data assimilation experiments in section 3.2

|  | overland flows | observing network |
|---|---|---|
| noOF_NoDA | none | no data assimilation |
| noOF_DA_obs1 | none | Figure 7a |
| noOF_DA_obs9 | none | Figure 7b |
| noOF_DA_obs361 | none | Figure 7c |
| OF_NoDA | simulated | no data assimilation |
| OF_DA_obs1 | simulated | Figure 7a |
| OF_DA_obs9 | simulated | Figure 7b |
| OF_DA_obs361 | simulated | Figure 7c |






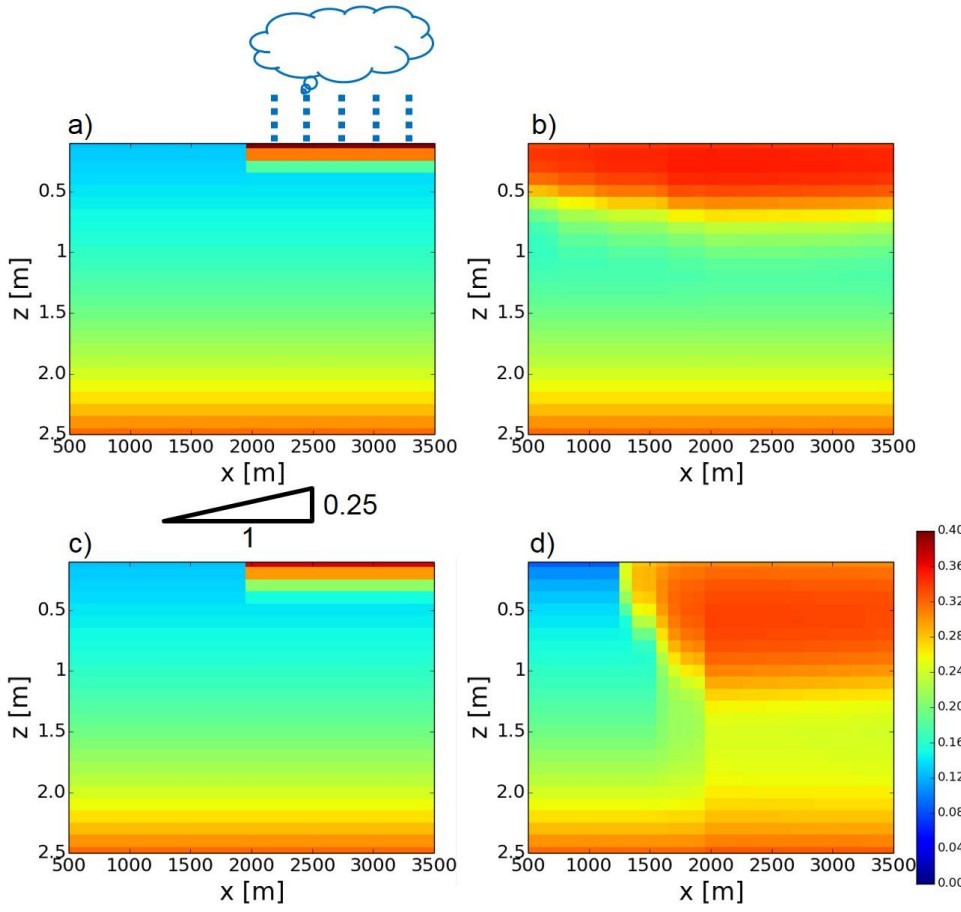


**Figure 1.** Distributions of volumetric soil moisture simulated by the synthetic reference runs. (a) The

distribution of volumetric soil moisture [m³/m³] simulated by the LOW_K synthetic reference run at t = 0h.

The schematic of the configuration of the synthetic reference runs is also shown (see also section 3). (b) same

as (a) but at t = 130h. (c,d) same as (a,b) but for the HIGH_K synthetic reference run.






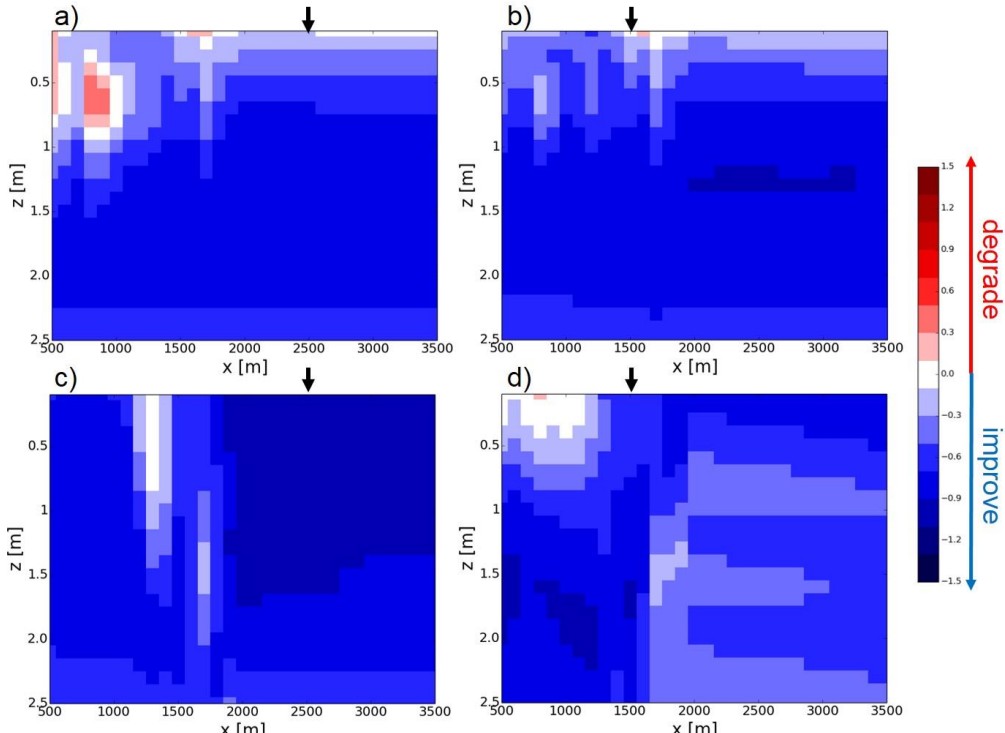


**Figure 2.** The improvement rates of the (a) LOW_K-UP_O, (b) LOW_K-DOWN_O, (c) HIGH_K_UP_O,

(d) HIGH_K-DOWN_O experiments (see Table 1 and section 3). Black arrows show the locations of the soil

moisture observations in each experiment.




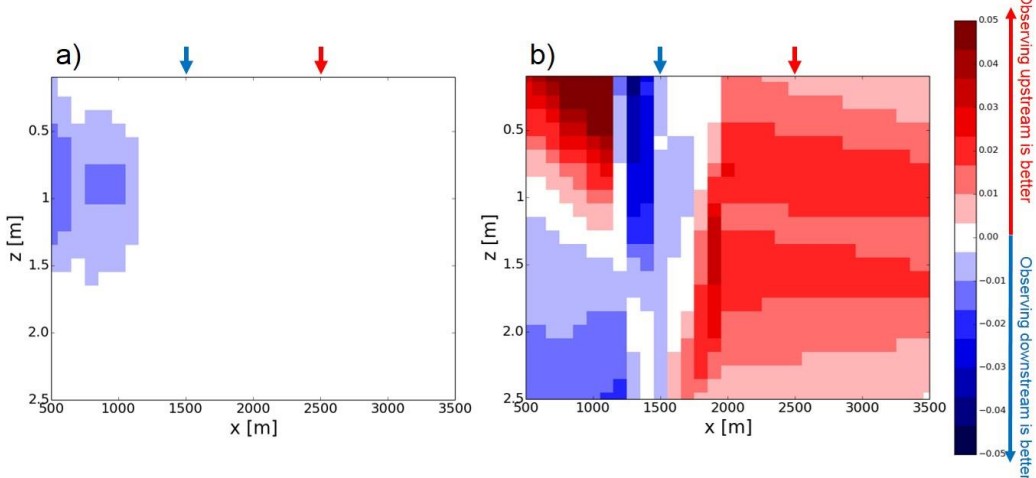


**Figure 3.** (a) The difference of time-mean RMSEs between the LOW_K-UP_O and LOW_K-DOWN_O

experiments (see Table 1 and section 3). Red (blue) color indicates that the observations in the upper (lower)

part of the slope reduce time-mean RMSE by data assimilation better than those in the lower (upper) part of

the slope (see also arrows which are the locations of the observations). (b) same as (a) but for the difference

between the HIGH_K-UP_O and HIGH_K-DOWN_O experiments.






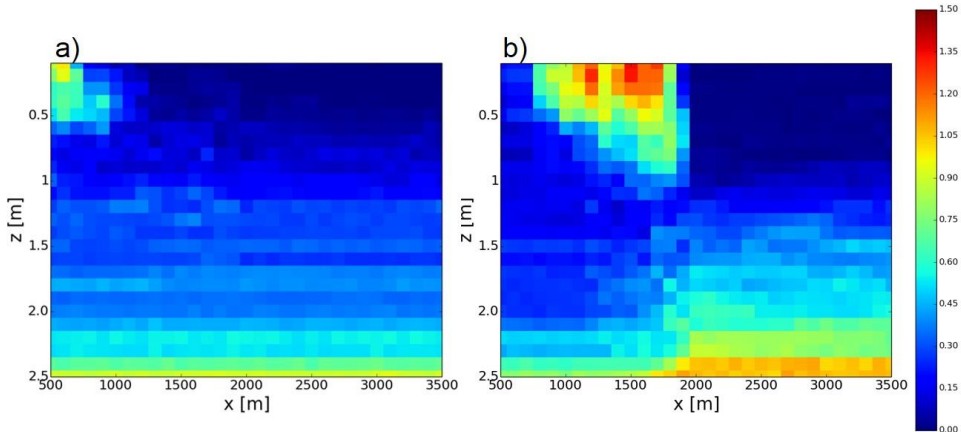


**Figure 4**. The Kullback-Leibler divergence of the NoDA experiment generated by (a) the LOW_K reference

and (b) the HIGH_K reference at t = 130h (see also Figure 1b and 1d).


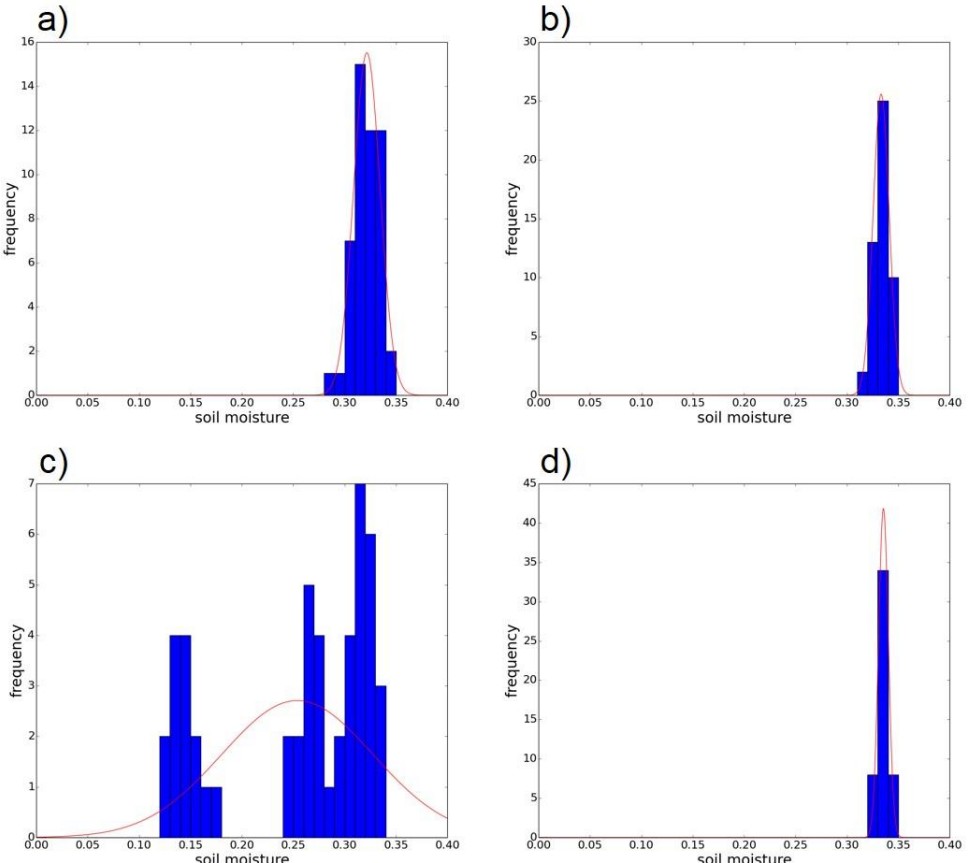


**Figure 5**. (a) The histogram (blue bars) of the volumetric soil moisture simulated by the NoDA experiment
(see section 3) with the LOW_K reference at x=1500m, z=0.5m, and t=130h (see also Figure 4). Red line
shows the Gaussian distribution with the mean and variance sampled by the ensemble. (b) same as (a) but at
x=2500m, z=0.5m, and t=130h. (c) same as (a) but for the HIGH_K reference. (d) same as (c) but at x=2500m,
z=0.5m, and t=130h.






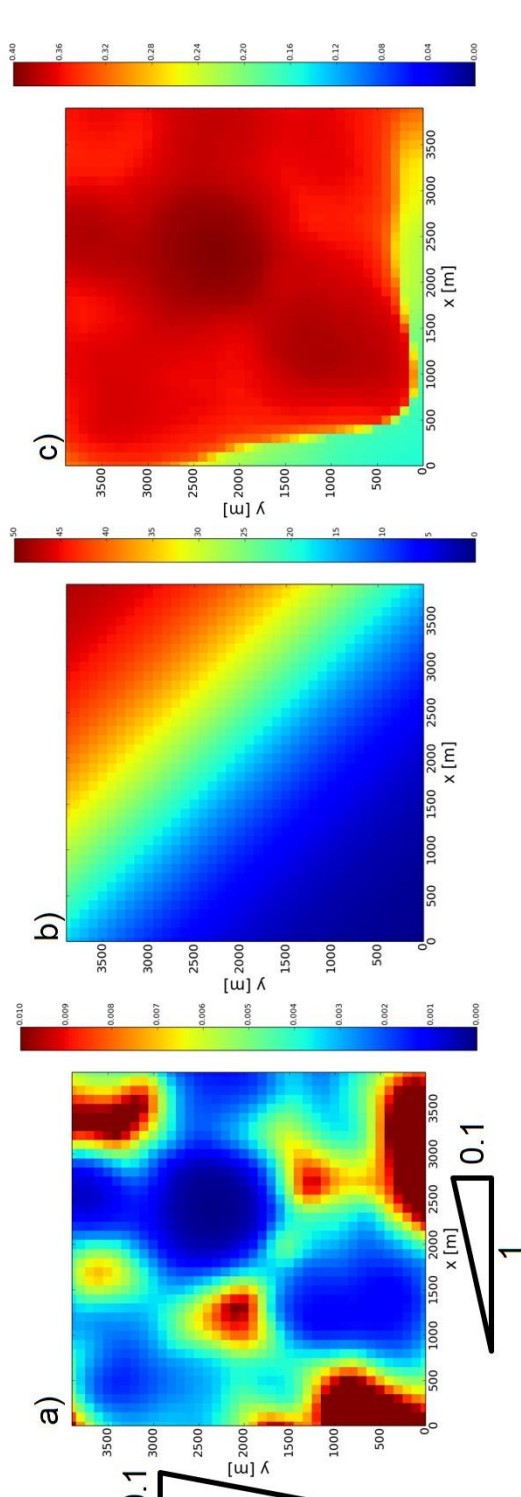


**Figure 6.** (a) Distribution of surface saturated hydraulic conductivity [m/h] in the synthetic reference. (b) Distribution of rainfall rate [mm/h] in the synthetic
reference. (c) Surface volumetric soil moisture [m³/m³] at t = 5 [h] in the synthetic reference.





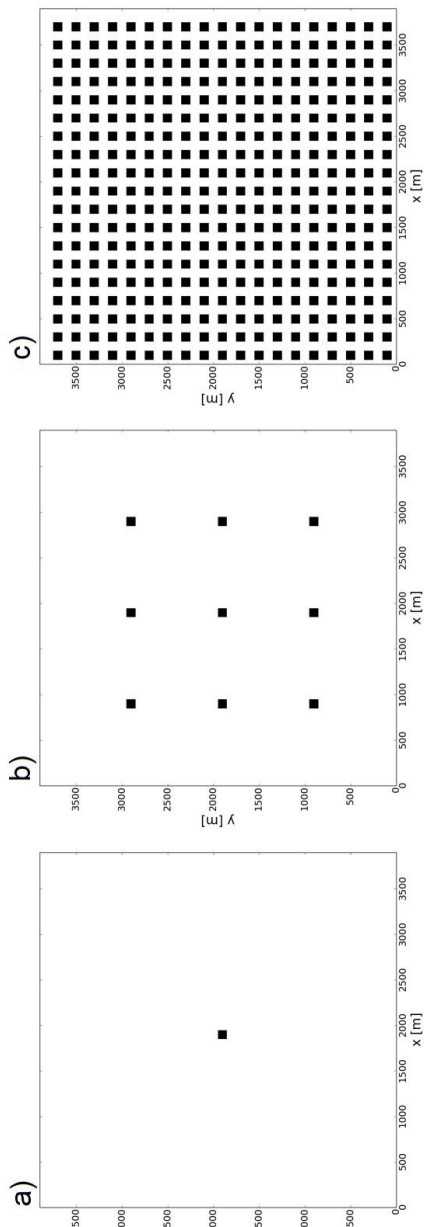

**Figure 7.** Observing networks. Black boxes are observed grids. (a) obs1, (b) obs9, (c) obs361 See also section 3.2.1.







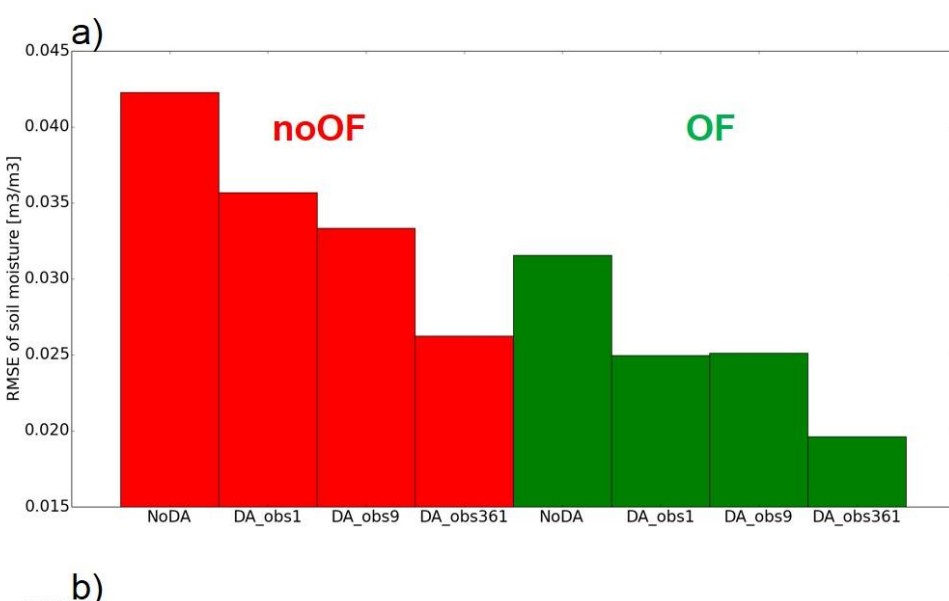

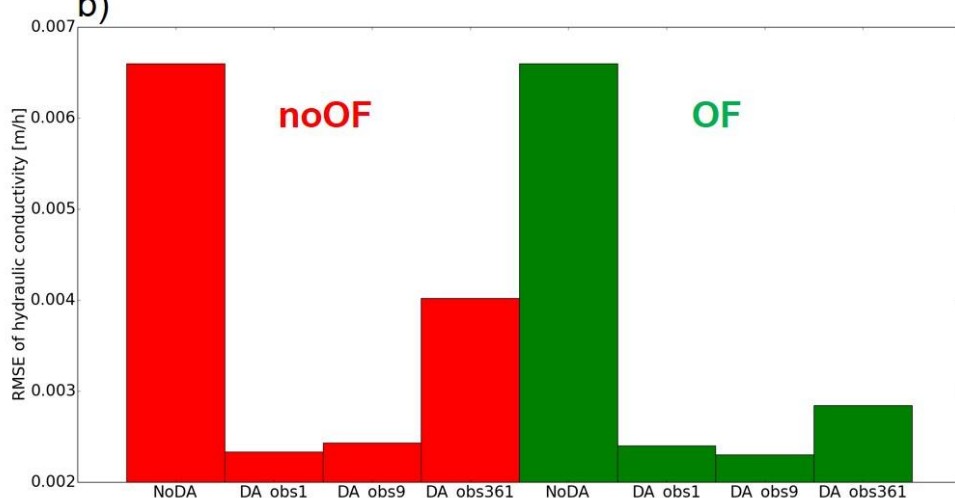


**Figure 8.** Time-mean RMSEs of the estimation of (a) soil moisture and (b) hydraulic conductivity. Red and

green bars are results of the noOF and OF configuration, respectively (see section 3.2.1 and Table 2).




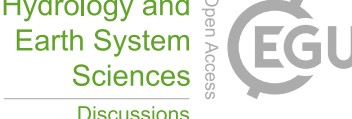

**Figure 9.** Differences of time-mean soil moisture RMSEs between the DA experiments and the OF_NoDA experiment. (a) OF_DA_obs1, (b) OF_DA_obs9 (c) OF_DA_obs361 (d) noOF_DA_obs1, (e) noOF_DA_obs9, (f) noOF_DA_obs361.






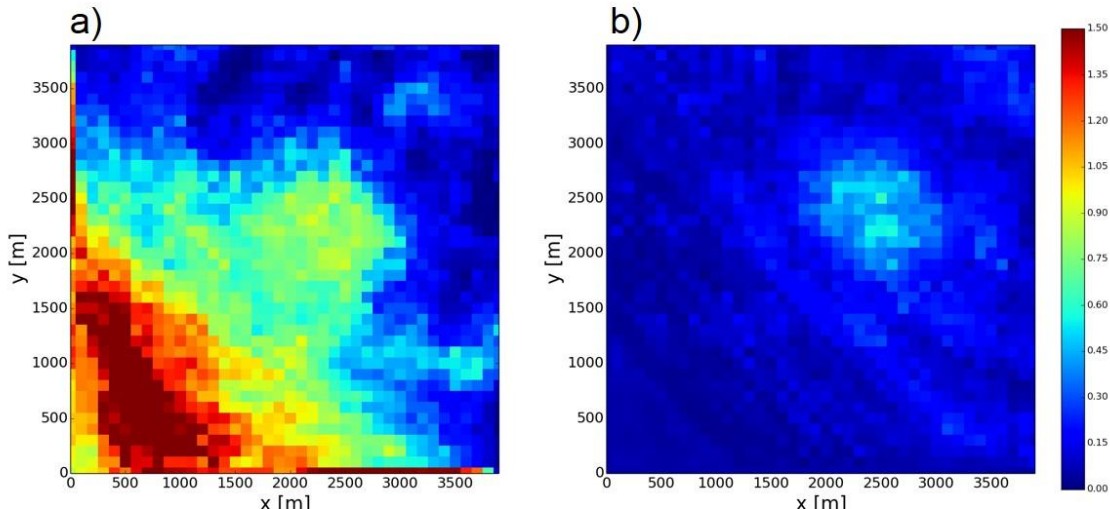


**Figure 10.** The Kullback-Leibler divergence of ensemble members generated by the (a) OF_NoDA and (b)
noOF_NoDA experiments at t = 4 [h].







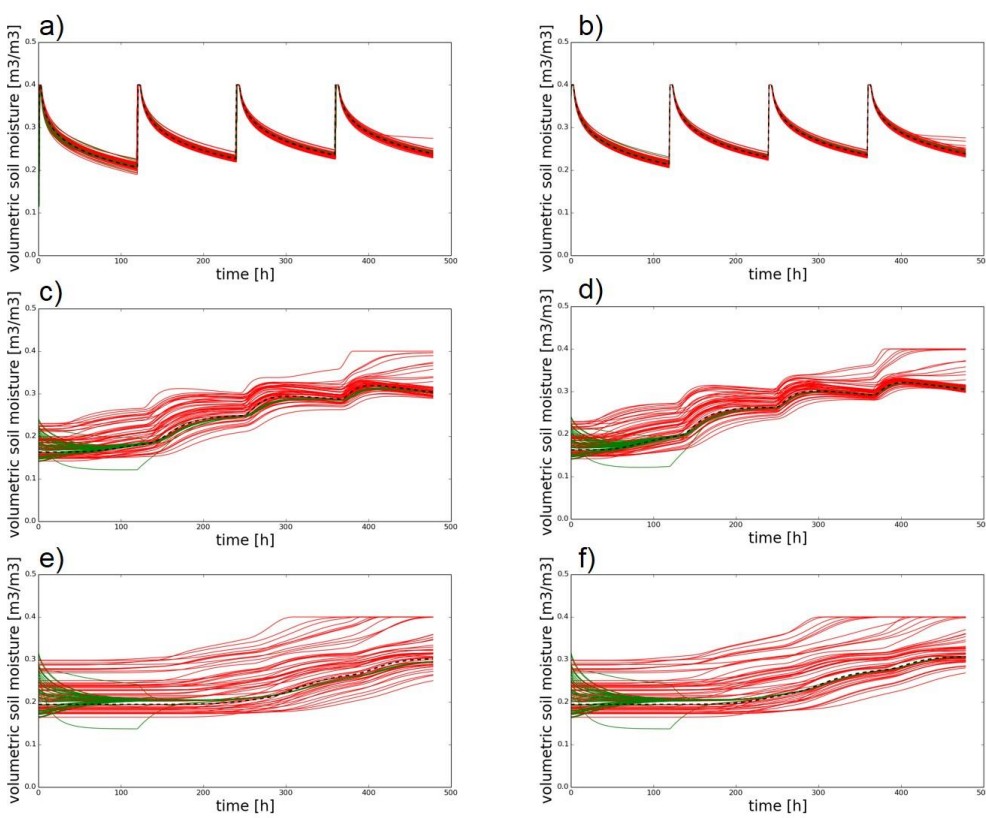


**Figure S1.** Time series of volumetric soil moisture simulated by the synthetic reference run (black dashed

line), the NoDA experiment (red lines), and the DA experiment (green lines) in the LOW_K-UP_O experiment

at a) x=1500m, z=0.05m; (b) x=2500m, z=0.05m; c) x=1500m, z=1.0m; (d) x=2500m, z=1.0m; e) x=1500m,

z=1.5m; (f) x=2500m, z=1.5m. In the DA experiment, initial guesses are used for this figure.






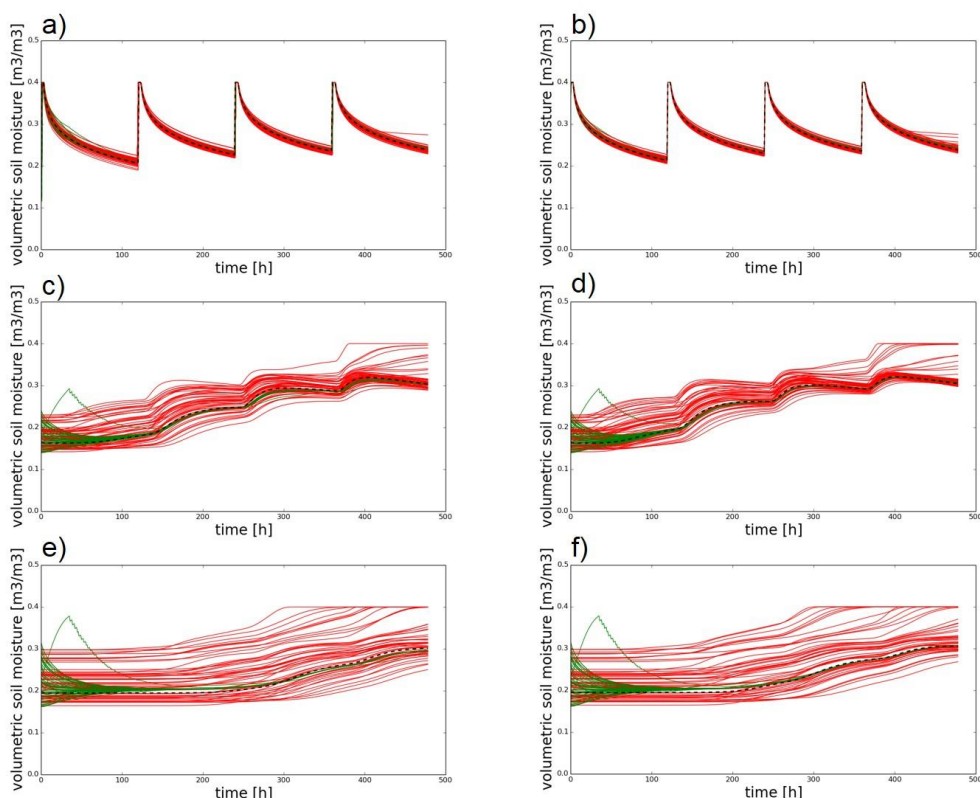


**Figure S2.** Same as Figure S1 but for the LOW_K-DOWN_O experiment.





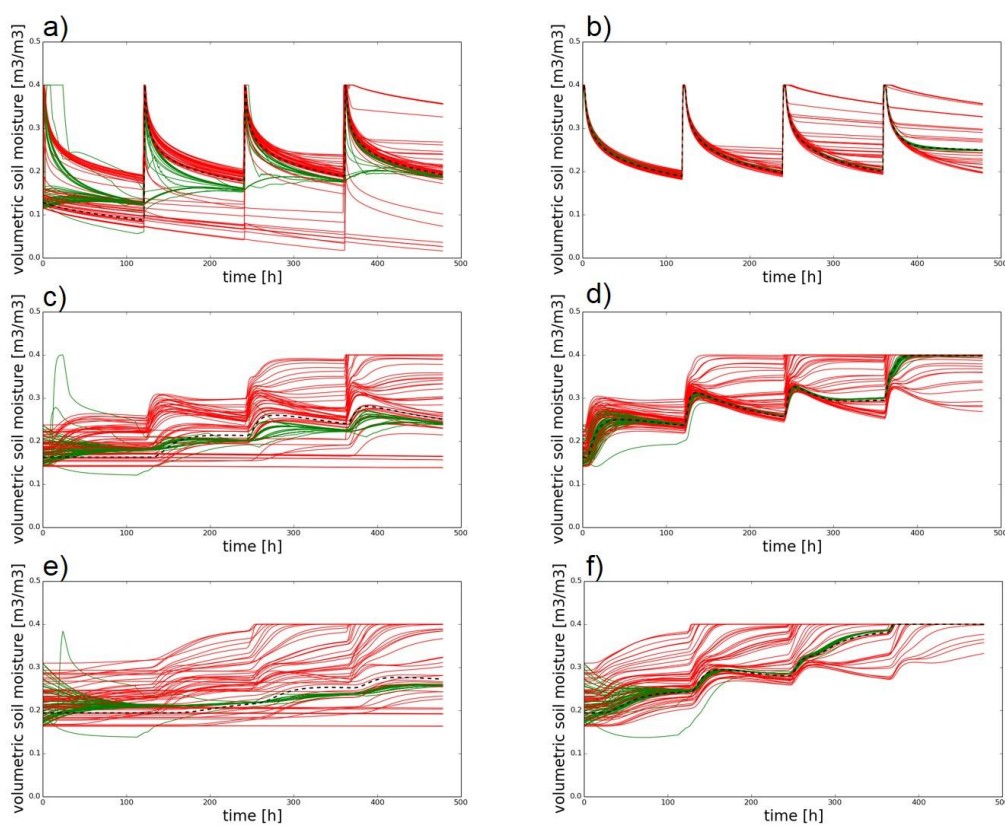


**Figure S3.** Same as Figure S1 but for the HIGH_K-UP_O experiment.




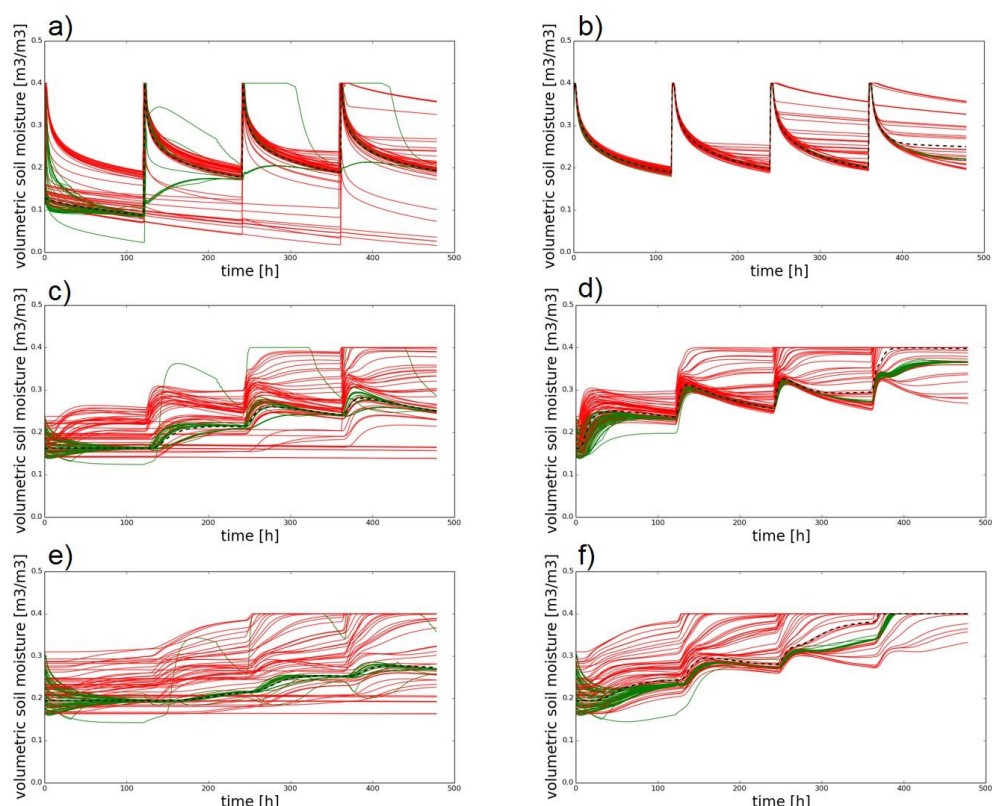


**Figure S4.** Same as Figure S1 but for the HIGH_K-DOWN_O experiment.






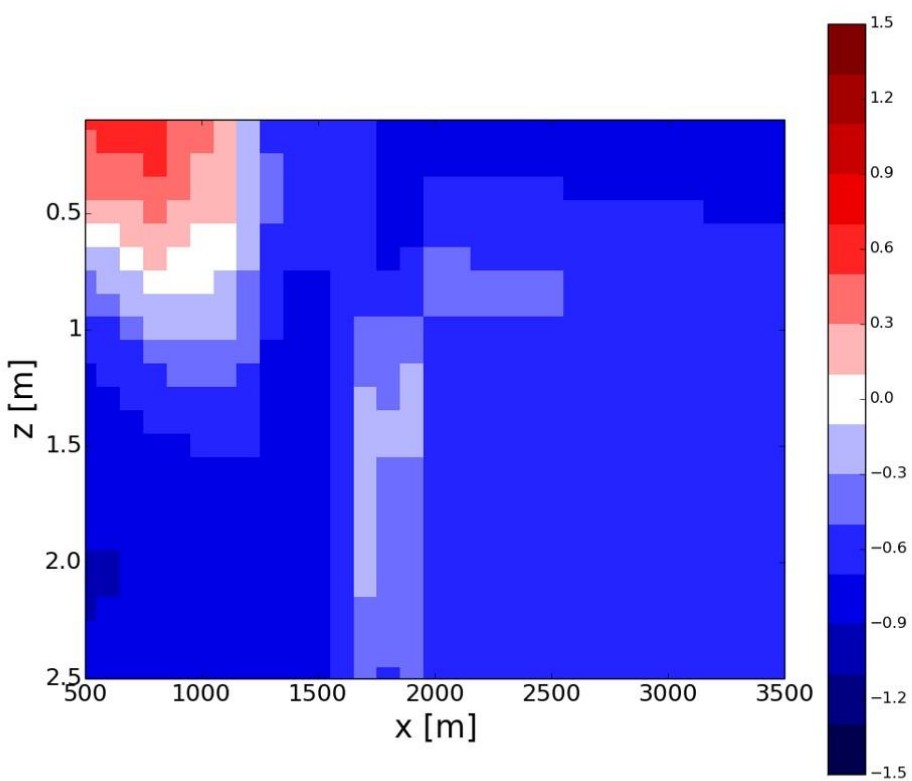


**Figure S5.** The improvement rates of the HIGH_K-DOWN_O experiment without a parameter optimization.






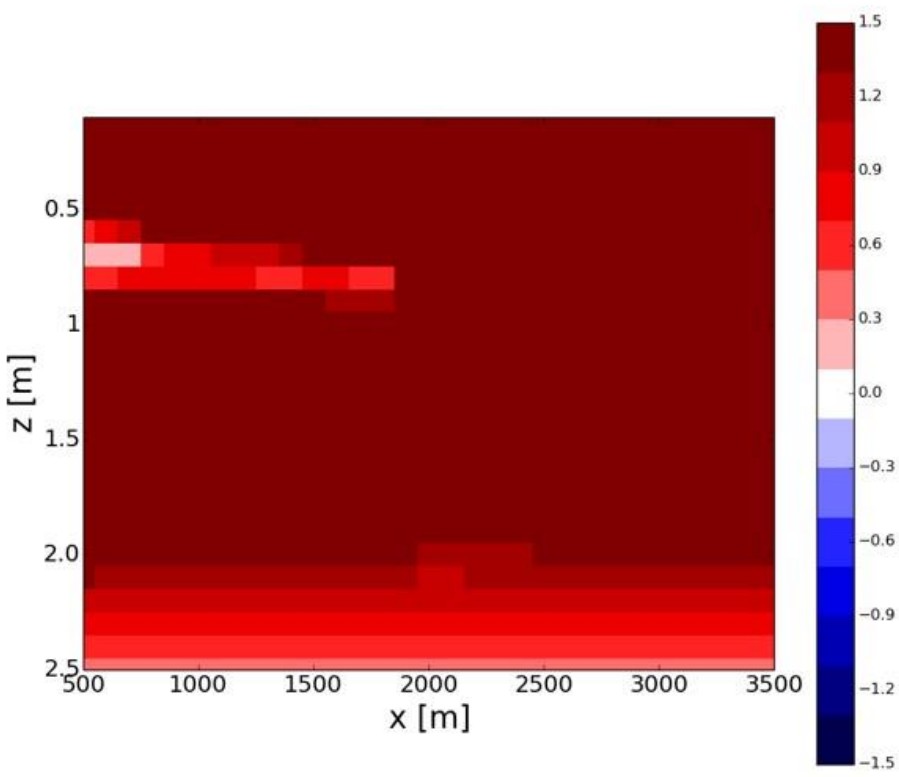


**Figure S6**. The improvement rates of the LOW_K-DOWN_O experiment where the topography-driven

surface flow is neglected in ParFlow.


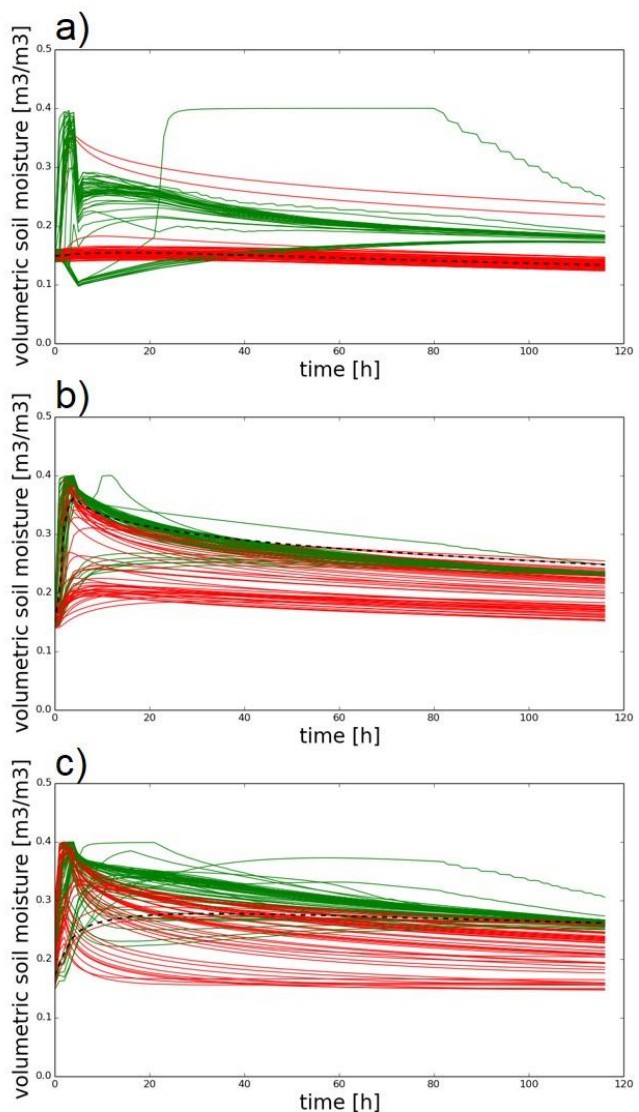


**Figure S7.** Time series of volumetric soil moisture simulated by the synthetic reference run (black dashed

line), the OF_NoDA experiment (red lines), and the OF_DA_obs361 experiment (green lines) at a) x=200m,

y=200m. z=0.15m; b) x=1200m, y=1200m, z=0.15m; c) x=2200m, y=2200m, z=0.15m.
