# Peer review of "Do surface lateral flows matter for data assimilation of soil moisture observations into hyperresolution land models? Running title: HYPERRESOLUTION LAND DATA ASSIMILATION"

_Hydrology and Earth System Sciences, 2019_

## Referee Comment (RC1) · Anonymous Referee #1 · 30 Sep 2019

The author of this paper used a synthetic case and indicated that topography-driven lateral surface flows induced by heavy rainfalls do matter for data assimilation of hydrological observations into hyper resolution land models. Although this paper reads well and the author provided a long discussion on results, these results are only based on a few deterministic measures, the author needs to clarify more detail and use additional matrices to evaluate his results. All the figures and tables are appropriate. This manuscript can be considered for publication after carefully addressing all of my concerns.

Minor Lines 62-63: ". . .by the data assimilation of microwave brightness temperature observations. . ." should be ". . .by assimilating microwave brightness temperature observations. . ."

Major: Line 203-206: please use some mathematical relationship to elaborate more what is van Genuchten relationship and how it has been used as H operator to convert pressure head to soil moisture. Parflow does not estimate the soil moisture directly? Lines 210-217: why did you use this approach to identify the closeness of the two PDFs, this seems a very old technique. It would have been much better before using each method you had explained the reason and necessity of using that approach. As this is the synthetic case and you are generating the pressure head and soil moisture observation accordingly, I am not sure how this study can be done on a real-case problem, which is very important, as its result would be more convincing. The author used only a few deterministic measures (e.g., RMSE) to assess the performance of the DA for all the assimilation scenarios in this study. Speaking of uncertainty quantification, both probabilistic and deterministic measures should be used to evaluate the effectiveness and usefulness of the EnKF model. These metrics although show how the simulated quantities could accurately match the observations, it does not provide any insight on the reliability of the predicted values. Therefore, I recommend using the following paper, in which the authors provided a comprehensive description of different probabilistic performance measures, such as Reliability and 95% exceedance ratio (ER95). These measures have been extensively used in many studies to evaluate the quality of the posterior distribution. Abbaszadeh, P., Moradkhani, H., & Daescu, D. N. (2019). The Quest for Model Uncertainty Quantification: A Hybrid Ensemble and Variational Data Assimilation Framework. Water Resources Research,55, 2407–2431. https://doi.org/10.1029/2018WR023629.

Lines 622 and 623: "Although particle filtering in a high dimensional system suffers from the "curse of dimensionality", please highlight that this can be resolved through improvements of importance sampling in PFs, and therefore it provides the potential for data assimilation application in large-scale systems" for more discussion the

readers can be referred to the following papers: P. Van Leeuwen. (2009). Particle Filtering in Geophysical Systems. Mon. Weather Rev., 137 (12), pp. 4089-4114. https://doi.org/10.1175/2009MWR2835.1 Lines 633-634- How do you convince that this "In addition, in the virtual experiment of this paper, I neglected some of the important land processes such 634 as transpiration, canopy interception, snow, and frozen soil." is a correct pre-assumptions.

---

## Referee Comment (RC2) · Anonymous Referee #2 · 6 Jan 2020

Journal: HESS Title: Do surface lateral flows matter for data assimilation of soil moisture observations into hyperresolution land models?

Author(s): Yohei Sawada MS No.: hess-2019-324 MS Type: Research Article

General Comments:

The author discusses the effects of inclusion of lateral transfer on propagation of soil moisture and its assimilation in a land surface model.

The importance of lateral transfers for land surface modeling is an important and open questions in the field, and hence worth performing diagnostic experiments. In my

view, studying the Data Assimilation (DA) results when lateral transfers are included or ignored in a land surface model (as it is done in this paper) will highlight more the efficiency of the DA technique rather than the impacts of lateral flow representation. Therefore, I find it more informative to discuss how soil moisture updates in a DA scheme differ when lateral transfers are switched on and off in a land surface model. Because this will highlight the importance of the processes that we know they occur but are overlooked in land surface models.

Moreover, the author claims that the focus of the study is "How and when surface water flows driven by local topography matter for data assimilation of soil moisture observations- line 19-20" but he fails to answer these questions shortly in the abstract or throughout the manuscript.

In line 24-26, the author indicate that the non-gaussianity of the background error disturbs the DA technique. Then why the author pursue this method while the very assumption needed for the DA technique presented here is violated?

The abstract doesn't state clearly what are the point by point messages that the manuscript intend to deliver.

In general, the arguments in interpreting the results are weak.

Major concerns:

Line 24, state variable and parameter variable are two technical terms that need to be explained for none-DA-community.

Line 37, what are examples of conventional land surface models, at what spatial resolution do they operate? Doesn't it make more sense to design the synthetic experiments at the same spatial resolution of typical land surface models that you are referring to?

Line 39-44, the author report the existing hyper-resolution land surface models that include lateral transfer. Given that such set ups already exist, what is the need to set up a synthetic experiment with synthetic observations, etc?

Line 50-52: you mean the challenge of land data assimilation is to "improve" the unobservable variables "using" observations by propagating. . .. ? Moreover, explain state and parameter space and their differences.

Line 44-47, be more specific about the outcome of the studies rather than sticking to "play an important role in terrestrial water and energy fluxes".

Line 66, for people outside the DA-community you need to explain what state variable, forcing variable and output variable are.

Line 87-89, what was the main outcome?

Line 87-91: Do these model set up include lateral transfers between the grid-cells? (if they have solved 3D Richards equation). In that case, what is new in synthetic experimental design of your study that have not been tested before?

Line 133, describe the van Gnuchten relationships you are refereeing to.

Line 146, this is an uncommon way of representing an equation. You may separate the two equations for x and Y as presented in Eq. 3 of Kollet and Maxwell 2006.

Line 149-150: summarize what are those methodology and numerical methods that are used to solve the equations. You are asking the readers to read 6 other papers to get the basic information about your simulations.

Section 2. Method, line 109-150: the description of the model that is used is not informative and comprehensive. Consider restructuring and adding more details in such way that it covers the following: 1) how the unsaturated zone is modelled? 2) How surface flows are modeled? 3) How the coupling of the two is being done? 4) What is the numerical approach taken for solving the equations?

Section 2.2. Data Assimilation, line 153-206: Please explain the topic of DA in a plain language and for audiences larger than merely DA-community. What is the general philosophy there? Why Kalman filter? Why Ensemble Transfer Kalman Filter? Is it

more accurate? is it faster? Please justify the choices you made.

Line 193-194: what do you mean? Please explain what an "ensemble inflation method" is. Explain how this method correct for this?

Line 195: again you are throwing a term and you are asking your audience to read an entire paper to understand what "relaxation to prior perturbation method (RTPP)" is!

Line 199: where does this number come from? Report the literature or justify your choice.

Line 201-206: Badly written. It is not clear which observations are assimilated in the system. It is not clear what are the state variables that are updated in every modeling time step.

Line 210-217: Please better explain what kind of measure "KLD" is, what it measures and why you used this measure to quantify the non-gaussianity of the background error? What do you do with the asymmetry of KLD? Is P Gaussian distribution or q?

Line 235: where did you get these estimates? Provide references or justify your choice.

Section 3.1.1 experimental design: what do you want to achieve with these two synthetic experimental design? Please explain why you made your choices of experimental design more clearly. e.g., Why dividing your domain to rain and no-rain regions?

Line 268-269: most land surface models use satellite observations of surface soil moisture for DA. Observations are usually not available to the depth of 1m! Please explain more clearly what you want to gain by assimilating SM at several depth?

How reference runs are different from no-assimilation runs?

Line 292: What is LDAS?

Line 322: What do you mean by Ks,surface is accurate?

Line 322: the increase of RSME or IR index?

Line 324-326: check the structure of the sentence.

Line 325: RMSE of what variable?

Line 329-330: this statement is not correct!

Line 332: where does this optimized value for K_sat come from? How do you infer its accuracy?

Line 335-336: Please elaborate what this difference indicates.

Line 342: How do you explain figure 2a then? The valley bottom in figure 2a (left corner) should theoretically improve when lateral transfers are included (when the uphill is raining). This figure shows that the observations of SM (also in depth) didn't improve the predictions downhill although lateral transfer processes are included in your model!

Line 356-363: I am confused! How did you conclude this? Didn't you assimilate SM observations across all areas (where it rains and where it does not rain)? The DA results are not sufficiently and clearly explained.

Line 365-366: what do you mean?

Line 370-371: yes, because of higher Hydraulic conductivity!

Line 395: nonlinear dynamics of what?

Line 409: you don't show that the errors are correlated.

Line 412-413: This is what most land surface models do when they assimilate SM and they still show improvements!

Line 435: why is this constant number added? Please justify the choices you made.

Line 467: what is "OF"?

Line 488-492: weak justification. What is over fitting? what is wrong with that? why does it have a negative impact? elaborate.. also in lines 505-507.

Line 522-525: what variable are you talking about?

Line 533-535: then this assimilation technique is not suitable for topographically driven systems.

Line 553, add a reference to this argument.

Line 564-566: knowing that Gaussian distribution of background error is a pre-requisite, why this method then?

Line 594: what is localization method?

Bottom line: I don't understand the choice of experimental set up, the data assimilation technique chosen given the non-gaussianity of the background error, and the justification of the simulation results.

Minor comments:

Consider breaking long sentences into multiple sentences. Examples of this include: Line 26-30, 55-61, 69-74

Line 111, what do you mean by "parallel simulation platform"?

Line 113, provide examples of such land models.

Line 160-171: what is Pa? In general, what do "a" and "f" stand for?

Line 647: space after error

---

## Author Comment (AC1) · 31 Jan 2020

Response letter of hess-2019-324-RC1

Please find the responses to the comments.

Comments made by the reviewer were highly insightful. They allowed me to greatly improve the quality of the manuscript. I described the response to the comments.

Each comment made by the reviewers is written in *italic* font. I numbered each comment as (n.m) in which n is the reviewer number and m is the comment number. In the revised manuscript, changes are highlighted in yellow.

I trust that the revisions and responses are sufficient for my manuscript to be published in *Hydrology and Earth System Sciences*

**Responses to the comments of Reviewer #1**

*The author of this paper used a synthetic case and indicated that topography-driven lateral surface flows induced by heavy rainfalls do matter for data assimilation of hydrological observations into hyper resolution land models. Although this paper reads well and the author provided a long discussion on results, these results are only based on a few deterministic measures, the author needs to clarify more detail and use additional matrices to evaluate his results. All the figures and tables are appropriate.*

*This manuscript can be considered for publication after carefully addressing all of my concerns.*

*(1.1) Minor Lines 62-63: "...by the data assimilation of microwave brightness tempera-ture observations..." should be "...by assimilating microwave brightness temperature observations..."*

→ I have modified it following the reviewer's instructions.

"Sawada et al. (2015) successfully improved the simulation of root-zone soil moisture by ==assimilating== microwave brightness temperature observations which include the information of vegetation water content."

*(1.2) Major: Line 203-206: please use some mathematical relationship to elaborate more what is van Genuchten relationship and how it has been used as H operator to convert pressure head to soil moisture. Parflow does not estimate the soil moisture directly?*

→ I have clarified the van Genuchten relationship in the revised version of the paper.

$$S_W(h) = \frac{S_{sat} - S_{res}}{(1 + (\alpha h)^n)^{\left(1 - \frac{1}{n}\right)}} + S_{res} \qquad (4)$$

$$k_r(h) = \frac{\left(1 - \frac{(\alpha h)^{n-1}}{(1 + (\alpha h)^n)^{\left(1 - \frac{1}{n}\right)}}\right)^2}{(1 + (\alpha h)^n)^{\frac{\left(1 - \frac{1}{n}\right)}{2}}} \qquad (5)$$

where $\alpha$ [L-1] and n [-] are soil parameters, $S_{sat}$ is the relative saturated water content and $S_{res}$ is the relative residual saturation.

Yes, Parflow does estimate the soil moisture directly so that I did not need to formulate the complicated H for data assimilation. What I wanted to say here is that I directly adjusted pressure head by assimilating (synthetic) volumetric soil moisture observation. The assimilated observation variables are not consistent to the adjusted state variables. Therefore, in the calculation of background covariance (equations (10) and (11)), the van Genuchten relationship can be recognized as H although

I did not need the van Genuchten relationship in data assimilation since volumetric soil moisture has already been calculated by Parflow. This point was indeed unclear in the original version of the paper and I have clarified this issue in the revised version of the paper.

> "I assimilated volumetric soil moisture observations so that $y^f$ and $y^o$ are simulated and observed volumetric soil moisture, respectively. The van Genuchten relationship converts the adjusted state variables $x^f$ to the observable variables $y^f$ and can be recognized as an observation operator $\mathcal{H}$. However, since volumetric soil moisture $y^f$ has already been calculated by Parflow, I did not need the van Genuchten relationship in data assimilation."

*(1.3) Lines 210-217: why did you use this approach to identify the closeness of the two PDFs, this seems a very old technique. It would have been much better before using each method you had explained the reason and necessity of using that approach. As this is the synthetic case and you are generating the pressure head and soil moisture observation accordingly, I am not sure how this study can be done on a real-case problem, which is very important, as its result would be more convincing. The author used only a few deterministic measures (e.g., RMSE) to assess the performance of the DA for all the assimilation scenarios in this study. Speaking of uncertainty quantification, both probabilistic and deterministic measures should be used to evaluate the effectiveness and usefulness of the EnKF model. These metrics although show how the simulated quantities could accurately match the observations, it does not provide any insight on the reliability of the predicted values. Therefore, I recommend using the following paper, in which the authors provided a comprehensive description of different probabilistic performance measures, such as Reliability and 95% exceedance ratio (ER95). These measures have been extensively used in many studies to evaluate the quality of the posterior distribution. Abbaszadeh, P., Moradkhani, H., & Daescu, D. N. (2019). The Quest for Model Uncertainty Quantification: A Hybrid Ensemble and Variational Data Assimilation Framework. Water Resources Research,55, 2407–2431. https://doi.org/10.1029/2018WR023629.*

→ Please note that I did not use the KLD as an evaluation metrics although the reviewer provided this comment in the context of evaluation metrics. Although the KLD is old, it is widely used in the context of machine learning. The KLD has also been used to measure the Gaussianity in the context of data assimilation so that I used it to evaluate how the ensemble simulation follows the Gaussian distribution. This point was indeed unclear in the original version of the paper. I have clarified this point in the revised version of the paper. This modification includes the response to the other reviewer's comment.

> "To evaluate the non-Gaussianity of the background error sampled by an ensemble, I used the Kullback-Leibler divergence (KLD) (Kullback and Leibler 1951):
>
> $$D_{KL}(p, q) = \sum_i p(i) log \frac{p(i)}{q(i)} \quad (13)$$

where $D_{KL}(p, q)$ is the KLD between two probabilistic distribution functions (PDFs), $p$ and $q$. If two PDFs are equal for all $i$, $D_{KL}(p, q) = 0$. A large value for $D_{KL}(p, q)$ indicates that the two PDFs, $p$ and $q$, substantially differ from each other. Therefore, the KLD can be used as an index to evaluate the closeness of two PDFs. In this study, I compared the PDF of the ensemble simulation (p in equation (13)) with the Gaussian PDF which has the mean and variance of the ensembles (q in equation (13)). A large value for $D_{KL}(p, q)$ indicates the state variables simulated by ensembles do not follow the Gaussian PDF. It should be noted that the KLD is not symmetric $(D_{KL}(p, q) \neq D_{KL}(q, p))$. The KLD has been used to quantitatively evaluate the Gaussianity of the sampled background error in the studies on data assimilation (e.g., Kondo and Miyoshi 2019; Duc and Saito 2018)."

The reviewer suggested clarifying the reason of the choice of methodology. Generally, I chose ParFlow and EnKF because they are widely accepted in the community. I would like to clarify how surface lateral flows matter in the widely accepted methodology. For Parflow, this point has already been clarified in the original version of the paper. I additionally emphasized this point in the revised version of the paper (the response to the other reviewer's comment is also included below):

"ParFlow is an open source platform which realizes fully integrated surface-groundwater flow modeling (Kollet and Maxwell 2006; Maxwell et al. 2015). This model can be efficiently parallelized in high performance computers and has been widely used as a core hydrological module in hyperresolution land models (e.g., Maxwell and Kollet 2008; Maxwell and Condon 2016; Fang et al. 2017; Kurtz et al. 2016; Maxwell et al. 2011; Williams and Maxwell 2011; Shrestha et al. 2014). Since I used this widely adopted solver as is and added nothing new to the model physics, I described the method of ParFlow to simulate integrated surface-subsurface water flows briefly and omitted the details of numerical methods. The complete description of ParFlow can be found in Kollet and Maxwell (2006), Maxwell et al. (2015) and references therein."

The EnKF is also widely accepted as the data assimilation algorithm for hyper-resolution land models. This point was unclear in the original version of the paper. I have clarified this point in the revised version of the paper.

"In this paper, the ensemble Kalman filter (EnKF) was applied to assimilate soil moisture observations into ParFlow. The EnKF has widely been applied to hyper-resolution land models (e.g., Camporese et al. (2009); Camporese et al. (2010); Ridler et al. (2014); Zhang et al. (2015); Kurtz et al. (2016); Zhang et al. (2018)). I examine if surface lateral flows matter for data assimilation of soil moisture observations into hyperresolution land models using this widely adopted data assimilation method. "

The reviewer suggested clarifying how to convince what I found here by real world applications. I believe that I could perform the similar experiment using in-situ soil moisture observations in the intensively observed river basins or using high resolution satellite observation. I have clarified this point at the end of the revised paper:

"Future work will focus on the real-world applications using intense in-situ soil moisture observation networks and/or high-resolution satellite soil moisture observations."

The reviewer also suggested using the probabilistic measures to evaluate the performance. Figure R1 shows the spatial distributions of 95% exceedance ratio (ER95) in the HIGH_K-DOWN_O experiment. In the NoDA experiment, ER95 is 0% everywhere. Since I assumed the large uncertainty in rainfall and saturated hydraulic conductivity and it is not mitigated in the NoDA experiment, the forecast is completely underconfident. Data assimilation made this too large ensemble spread smaller. However, the spatial averaged ER95 is 31% and much larger than 5% so that the ensemble forecast in the DA experiment is overconfident. This is probably because the number of rainfall events and/or the frequency of rainfall events is small (see Figure S1 and Figures of Abbaszadeh et al. (2019)). In hydrological models, rainfall events are the primary factor to increase the ensemble spread so that it is difficult to maintain the appropriate ensemble spread with the small number of rainfall events. Interestingly, the regions of good ER95 corresponds to the regions where RMSE is greatly reduced (please compare Figure R1b and Figure 2d) so that RMSE can be used as a good proxy of the probabilistic measure. Same conclusions can be obtained in the other synthetic experiments.

[Figure]

**Figure R1.** ER95 for (a) the NoDA experiment and (b) the DA experiment in the HIGH_K-DOWN_O setting.

I would like to propose not to include Figure R1 and the detailed discussion of the evaluation by the probabilistic measure although I briefly mentioned the importance of the probabilistic measure in the revised version of the paper. First, as the other reviewer revealed, this theoretical paper has already been very complicated for readers outside the community of theoretical and hydrologic data assimilation. To get many potential readers, I believe that I should not further add the results and figures if it is not absolutely necessary. Please note that the uncertainty quantification is the quite advanced topic. To my best knowledge, the probabilistic measures that the reviewer raised have been used mainly in the data assimilation of lumped and conceptual hydrologic models (e.g., Abbaszadeh et al. (2019)), which is the most matured research field in the hydrologic data assimilation. Currently, the studies on the data assimilation of hyper-resolution land models have not used these evaluation metrices. I believe that the take-home-message of this study can be described and validated without this probabilistic measure.

Second, the current experiment design was not appropriate to deeply discuss the evaluation by the probabilistic measures. As I discussed in Figure R1, in the synthetic experiment, the number of rainfall events is small, and the timing and magnitude of rainfall were not diversified. Therefore, I could not expect the enough amount of data to evaluate the long-term statistical property of the ensemble simulation as Abbaszadeh et al. (2019) did. This point is important to move on to the more realistic experiment implemented in the future. In the revised version of the paper, I have included this limitation in the discussion section citing Abbaszadeh et al. (2019).

> "The other limitation of this study is that I could not thoroughly evaluate the skill of the ensemble data assimilation to quantify the uncertainty of its prediction. Following Abbazadeh et al. (2019), I calculated the 95% exceedance ratio and found that the ensemble forecast was systematically overconfident (not shown). In the synthetic experiments of this study, the number of rainfall events was small, and the timing and magnitude of rainfall were not diversified. Due to this limited amount of data, it is difficult to deeply discuss the accuracy of the quantified uncertainty by data assimilation. While the skill of lumped hydrological models was often evaluated by the probabilistic performance measures such as the 95% exceedance ratio (e.g., Abbazadeh et al. (2019)), the uncertainty quantification of the simulation of hyper-resolution land models is in its infancy. How surface lateral flows affect the accuracy of the uncertainty quantification by data assimilation should be investigated using more realistic data."

*(1.4) Lines 622 and 623: "Although particle filtering in a high dimensional system suffers from the "curse of dimensionality", please highlight that this can be resolved through improvements of*

*importance sampling in PFs, and therefore it provides the potential for data assimilation application in large-scale systems" for more discussion the readers can be referred to the following papers: P. Van Leeuwen. (2009). Particle Filtering in Geophysical Systems. Mon. Weather Rev., 137 (12), pp. 4089-4114.* [https://doi.org/10.1175/2009MWR2835.1](https://doi.org/10.1175/2009MWR2835.1)

→ I fully agree with this reviewer's comment. Recently, some studies provided the methodological advances of PF although their applicability to hydrological models has not been discussed. Note that many of works cited by van Leeuwen et al. used conceptual models such as Lorenz96 and the applicability of these methodological advances to the real-world problems is still debated. I have included this issue in the revised version of the paper.

> "Although particle filtering in a high dimensional system suffers from the "curse of dimensionality" (e.g., Snyder et al. 2008), some studies developed the methodology to improve the efficiency of particle filtering (e.g., van Leeuwen 2009; Poterjoy et al. 2019)."

*(1.5) Lines 633-634- How do you convince that this "In addition, in the virtual experiment of this paper, I neglected some of the important land processes such as transpiration, canopy interception, snow, and frozen soil." is a correct pre-assumptions.*

→ First, these neglected processes can be modelled as a source term of ParFlow in many hyperresolution land models. Therefore, these processes do not modify the fundamental physical process simulated by ParFlow so that what I found in this study can be robust to the models which include these processes. This point was indeed unclear in the original version of the paper and I have clarified it in the revised version of the paper.

> "These processes affect the source term of equation (1) in hyper-resolution land models (e.g., Shrestha et al. 2014). Since the inclusion of the neglected processes do not change the structure of the original ParFlow, the findings of this study can be robust to the models which include these processes."

In addition, here I focused on the propagation of overland flows, whose timescale is relatively short compared with the neglected processes. Therefore, neglecting these processes may not have a large impact on the conclusion of this paper quantitatively. This point has already been discussed in the original version of the paper.

> "Although they are generally not primary factors in the propagation of overland flows generated by extreme rainfall, which has a shorter timescale than the neglected processes, those processes should be considered in the future."

---

## Author Comment (AC2) · 31 Jan 2020

Response letter of hess-2019-324-RC2

Please find the responses to the comments.

Comments made by the reviewer were highly insightful. They allowed me to greatly improve the quality of the manuscript. I described the response to the comments.

Each comment made by the reviewers is written in *italic* font. I numbered each comment as (n.m) in which n is the reviewer number and m is the comment number. In the revised manuscript, changes are highlighted in yellow.

I trust that the revisions and responses are sufficient for my manuscript to be published in *Hydrology and Earth System Sciences*

**Responses to the comments of Reviewer #2**

First of all, I strongly believe that most of the reviewer's concerns have already been addressed in the original version of the paper. However, I failed to accurately deliver what I've done to the reviewer due to the insufficient description of the original paper. I believe that the description of the paper has been significantly improved by the reviewer's comments. Please see my responses below.

*The author discusses the effects of inclusion of lateral transfer on propagation of soil moisture and its assimilation in a land surface model.*

*(2.1) The importance of lateral transfers for land surface modeling is an important and open questions in the field, and hence worth performing diagnostic experiments. In my view, studying the Data Assimilation (DA) results when lateral transfers are included or ignored in a land surface model (as it is done in this paper) will highlight more the efficiency of the DA technique rather than the impacts of lateral flow representation. Therefore, I find it more informative to discuss how soil moisture updates in a DA scheme differ when lateral transfers are switched on and off in a land surface model. Because this will highlight the importance of the processes that we know they occur but are overlooked in land surface models.*

→ In my understanding, the reviewer suggests discussing how soil moisture updates in a DA scheme differ when lateral transfers are switched on and off in a land surface model. I believe that this is exactly what I did in the whole paper especially in the section 3.2. Please see Figure S6 and the section 3.2. I have clarified that I did it in the abstract of the revised paper.

==I discuss how differently the ensemble Kalman filter works when surface lateral flows are switched on and off.==

*(2.2) Moreover, the author claims that the focus of the study is "How and when surface water flows driven by local topography matter for data assimilation of soil moisture observations- line 19-20" but he fails to answer these questions shortly in the abstract or throughout the manuscript.*

*In line 24-26, the author indicate that the non-gaussianity of the background error disturbs the DA technique. Then why the author pursue this method while the very assumption needed for the DA technique presented here is violated?*

→ I realized that I did not explicitly answer "when" and I did not intend to answer it (I can say surface lateral flows matter when surface lateral flows are generated by heavy rain, but it is too obvious). In the revised version of the paper, I deleted "when".

How do surface lateral flows matter for data assimilation? I believe that I answered the question in the latter part of the abstract.:

> "A horizontal background error covariance provided by overland flows is important to adjust the unobserved state and parameter variables. However, the non-Gaussianity of the background error provided by the nonlinearity of a topography-driven surface flow harms the performance of data assimilation. It is difficult to efficiently constrain model states at the edge of the area where the topography-driven surface flow reaches by linear-Gaussian filters, which brings the new challenge in land data assimilation for hyperresolution land models. This study highlights the importance of surface lateral flows in hydrological data assimilation."

Please note that I did not pursue EnKF. I did not propose to use EnKF. I am not the first person who used EnKF for data assimilation of the high-resolution hydrological model. Many previous studies cited in the introduction pursued EnKF and proposed to use EnKF probably because it is easy to implement and computationally efficient. In this paper, I discussed how this widely adopted EnKF works with surface lateral flows, which has been overlooked in the previous studies. As the reviewer indicated, I found the potential problems in EnKF and I would like to tell them to researchers in the community as soon as possible since EnKF is now being widely used. In the original paper, it is clarified that EnKF has been used by many previous works in the Introduction section.

> "Previous works successfully applied Ensemble Kalman Filters (EnKF) to 3-D Richards' equation-based integrated surface-groundwater models."

In the revised version of the paper, I have clarified that EnKF is a widely adopted method in the method section.

> "The EnKF has widely been applied to hyper-resolution land models (e.g., Camporese et al. (2009); Camporese et al. (2010); Ridler et al. (2014); Zhang et al. (2015); Kurtz et al. (2016); Zhang et al. (2018)). I examined if surface lateral flows matter for data assimilation of soil moisture observations into hyperresolution land models using this widely adopted data assimilation method."

Please note that I have some suggestions to overcome the limitation of the existing algorithm in the discussion section. I pointed out that the widely adopted method has limitations and proposed the future direction of the methodological development.

*(2.3) The abstract doesn't state clearly what are the point by point messages that the manuscript intend to deliver.*

→ Please see every response described below. It will make the take-home-message of this paper clearer. Then, please go back to the abstract and point out what should be done to improve it. I will be ready to further improve the abstract in the next iteration.

*(2.4) In general, the arguments in interpreting the results are weak.*

→ Please see my responses below.

*Major concerns:*

*(2.5) Line 24, state variable and parameter variable are two technical terms that need to be explained for none-DA-community.*

→ I believe that state variable and parameter are not technical terms in data assimilation. They are technical terms in modeling. In abstract, I briefly show that state variables are pressure head and soil moisture and parameters are saturated hydraulic conductivity.

> "A horizontal background error covariance provided by overland flows is important to adjust the unobserved state variables (pressure head and soil moisture) and parameters (saturated hydraulic conductivity)."

In section 2.2, I describe the definition of them by showing the discretized dynamic systems.

> "The Parflow model can be formulated as a discrete state-space dynamic system:
>
> $$x(t+1) = f\big(x(t), \theta, u(t)\big) + q(t) \qquad (8)$$
>
> where $x(t)$ is the state variables (i.e. pressure head), $\theta$ is the time-invariant model parameters (i.e. saturated hydraulic conductivity), $u(t)$ is the external forcing (i.e., rainfall and evapotranspiration), and $q(t)$ is the noise process which represents the model error. In data assimilation, it is useful to formulate an observation process as follows:
>
> $$y^f(t) = \mathcal{H}\big(x(t)\big) + r(t) \qquad (9)$$
>
> where $y^f(t)$ is the simulated observation, $\mathcal{H}$ is the observation operator which maps the model's state variables into the observable variables, and $r(t)$ is the noise process which represents the observation error. The purpose of EnKF (and any other data assimilation methods) is to find the optimal state variables $x(t)$ based on the simulation $y^f(t)$ and observation (defined as $y^o$) considering their errors ($q(t)$ and $r(t)$)"

*(2.6) Line 37, what are examples of conventional land surface models, at what spatial resolution do they operate? Doesn't it make more sense to design the synthetic experiments at the same spatial resolution of typical land surface models that you are referring to?*

→ I did not include the spatial resolution of the conventional land surface models. I have described this point in the revised version of the paper.

> "While conventional land surface models (LSMs) assume that lateral water flows are negligible at the coarse resolution (>25km) and solve vertical one-dimensional Richards equation for the soil moisture simulation (e.g., Sellers et al. 1996; Lawrence et al. 2011), currently proposed hyperresolution land models, which can be applied at a finer resolution (<1km), explicitly consider surface and subsurface lateral water flows (e.g., Maxwell and Miller 2005; Tian et al. 2012; Shrestha et al. 2014; Niu et al. 2014)."

The reviewer suggests designing the synthetic experiments at this coarse spatial resolution. In this spatial resolution, lateral flows are neglected or parameterized so that the water in one grid does not move to the adjacent grid. This is mainly because the hillslopes, drivers of lateral flows, cannot be practically resolved in this coarse spatial scale. Therefore, I could not answer my research question "Do surface lateral flows matter for data assimilation of soil moisture observations into hyperresolution land models?" at the coarse resolution of the conventional land surface model. Our focus was data assimilation in the hyperresolution land modeling in which lateral flows are explicitly modelled. This point has already been included in the original version of the paper.

> "While conventional land surface models (LSMs) assume that lateral water flows are negligible at the coarse resolution (>25km) and solve vertical one-dimensional Richards equation for the soil moisture simulation (e.g., Sellers et al. 1996; Lawrence et al. 2011), currently proposed hyperresolution land models, which can be applied at a finer resolution (<1km), explicitly consider surface and subsurface lateral water flows (e.g., Maxwell and Miller 2005; Tian et al. 2012; Shrestha et al. 2014; Niu et al. 2014)."

> "The fine horizontal resolution can resolve slopes, which are drivers of a lateral transport of water, and realize the fully integrated surface-groundwater modeling."

*(2.7) Line 39-44, the author report the existing hyper-resolution land surface models that include lateral transfer. Given that such set ups already exist, what is the need to set up a synthetic experiment with synthetic observations, etc?*

→ I could not understand what the reviewer implies in this comment. I did not add anything to the existing hyper-resolution land surface models. I evaluated how the widely adopted data assimilation method works with surface lateral water transport (see also the response to the comment (2.2)). To do

this, the synthetic experiment is absolutely necessary. I guess that I should discuss the importance of the synthetic experiment recognized in the DA-community more deeply. In the revised version of the paper, I have included this point and showed that the synthetic experiment was widely adopted in the community.

"In this study, I performed two synthetic experiments. In the synthetic experiments, I generated the synthetic truth of the state variables by driving ParFlow with the specified parameters and input data. Then the synthetic observations were generated by adding the Gaussian white noise to this synthetic truth. The performance of data assimilation was evaluated by comparing the estimated state and parameter values by ETKF with the synthetic truth. This synthetic experiment has been recognized as an important research method to analyze how data assimilation works (e.g., Moradkhani et al. 2005; Camporese et al. 2009; Vrugt et al. 2013; Kurtz et al. (2016); Sawada et al. 2018)"

*(2.8) Line 50-52: you mean the challenge of land data assimilation is to "improve" the unobservable variables "using" observations by propagating.... ? Moreover, explain state and parameter space and their differences.*

→ I have modified this sentence following the reviewer's instruction.

"The grand challenge of land data assimilation is to improve unobservable variables using observations by propagating observations' information into model's high dimensional state and parameter space."

I explained state and parameter briefly in the abstract and the method section.

"A horizontal background error covariance provided by overland flows is important to adjust the unobserved state variables (pressure head and soil moisture) and parameters (saturated hydraulic conductivity)."

"The Parflow model can be formulated as a discrete state-space dynamic system:

$$x(t+1) = f\big(x(t), \theta, u(t)\big) + q(t) \qquad (8)$$

where $x(t)$ is the state variables (i.e. pressure head), $\theta$ is the model parameters (i.e. saturated hydraulic conductivity), $u(t)$ is the external forcing (i.e., rainfall and evapotranspiration), and $q(t)$ is the noise process which represents the model error. In data assimilation, it is useful to formulate an observation process as follows:

$$y^f(t) = \mathcal{H}\big(x(t)\big) + r(t) \qquad (9)$$

where $y^f(t)$ is the simulated observation, $\mathcal{H}$ is the observation operator which maps the model's state variables into the observable variables, and $r(t)$ is the noise process which represents the observation error. The purpose of EnKF (and any other data assimilation methods)

is to find the optimal state variables $x(t)$ based on the simulation $y^f(t)$ and observation (defined as $y^o$) considering their errors ($q(t)$ and $r(t)$)"

See also the response to the comment (2.5). I strongly believe that state and parameter are widely used in the paper which uses hydrological models (not DA).

*(2.9) Line 44-47, be more specific about the outcome of the studies rather than sticking to "play an important role in terrestrial water and energy fluxes".*

→ I have included this point in the revised version of the paper.

"Previous works indicated that a lateral transport of water strongly controls latent heat flux and the partitioning of evapotranspiration into base soil evaporation and plant transpiration (e.g., Maxwell and Condon 2016; Ji et al. 2017; Fang et al. 2017). This effect of a lateral transport of water on land-atmosphere interactions has been recognized (e.g., Williams and Maxwell 2011; Keune et al. 2016).

*(2.10) Line 66, for people outside the DA-community you need to explain what state variable, forcing variable and output variable are.*

→ See the responses to the comments (2.5) and (2.8). I have formulated them in the method section. Once again, they are not technical terms in the DA-community. They are widely used in many kinds of studies on hydrological modeling.

"The Parflow model can be formulated as a discrete state-space dynamic system:

$$x(t+1) = f\big(x(t), \theta, u(t)\big) + q(t) \qquad (8)$$

where $x(t)$ is the state variables (i.e. pressure head), $\theta$ is the model parameters (i.e. saturated hydraulic conductivity), $u(t)$ is the external forcing (i.e., rainfall and evapotranspiration), and $q(t)$ is the noise process which represents the model error. In data assimilation, it is useful to formulate an observation process as follows:

$$y^f(t) = \mathcal{H}\big(x(t)\big) + r(t) \qquad (9)$$

where $y^f(t)$ is the simulated observation, $\mathcal{H}$ is the observation operator which maps the model's state variables into the observable variables, and $r(t)$ is the noise process which represents the observation error. The purpose of EnKF (and any other data assimilation methods) is to find the optimal state variables $x(t)$ based on the simulation $y^f(t)$ and observation (defined as $y^o$) considering their errors ($q(t)$ and $r(t)$)"

*(2.11) Line 87-89, what was the main outcome?*

→ I have clarified the achievement of Kurtz et al. (2016) in the revised version of the paper.

"Kurtz et al. (2016) coupled the Parallel Data Assimilation Framework (PDAF) (Nerger and Hiller 2013) with the Terrestrial System Modelling Framework (TerrSysMP) (Shrestha et al. 2014) and successfully estimate the spatial distribution of soil moisture and saturated hydraulic conductivity in the synthetic experiment (see also Zhang et al. (2018)). In addition, Kurtz et al. (2016) indicated that their EnKF approach is computationally efficient in high-performance computers."

*(2.12) Line 87-91: Do these model set up include lateral transfers between the grid-cells? (if they have solved 3D Richards equation). In that case, what is new in synthetic experimental design of your study that have not been tested before?*

→ Yes, they include lateral transfers between grid cells. I believe that the unsolved issue has been identified in the next paragraph of the original paper. How lateral transfers work in the data assimilation system has not been clarified.

"Although the data assimilation of hydrological observations into hyperresolution land models has been successfully implemented in the synthetic experiments, it is unclear how topography-driven surface lateral water flows matter for data assimilation of soil moisture observations. Previous studies on data assimilation with high resolution models mainly focused on assimilating groundwater observations (e.g., Ait-El-Fquih et al. 2016; Rasmussen et al. 2015; Hendricks-Franssen et al. 2008). There are some applications which focused on the observation of soil moisture and pressure head in shallow unsaturated soil layers. However, in those studies, topography-driven surface flow has not been considered in the experiment (Kurtz et al. 2016) or the role of them in assimilating observations into the hyperresolution land models has not been quantitatively discussed (Camporese et al. 2010; Camporese et al. 2009). This study aims at clarifying if surface lateral flows matter for data assimilation of soil moisture observations into hyperresolution land models by a minimalist numerical experiment. "

*(2.13) Line 133, describe the van Gnuchten relationships you are refereeing to.*

→ I have described it in the revised version of the paper.

$$"S_W(h) = \frac{S_{sat} - S_{res}}{(1 + (\alpha h)^n)^{(1 - \frac{1}{n})}} + S_{res} \qquad (4)$$

$$k_r(h) = \frac{(1 - \frac{(\alpha h)^{n-1}}{(1 + (\alpha h)^n)^{\left(1 - \frac{1}{n}\right)}})^2}{(1 + (\alpha h)^n)^{\frac{\left(1 - \frac{1}{n}\right)}{2}}} \qquad (5)$$

where $\alpha$ [L-1] and n [-] are soil parameters, $S_{sat}$ is the relative saturated water content and $S_{res}$ is the relative residual saturation."

*(2.14) Line 146, this is an uncommon way of representing an equation. You may separate the two equations for x and Y as presented in Eq. 3 of Kollet and Maxwell 2006.*

→ I have modified this point following the reviewer's instruction.

"$v_{sw}$ is the two-dimensional depth-averaged water flow velocity [LT$^{-1}$] and estimated by the Manning's law:

$$v_{sw,x} = \left(\frac{\sqrt{S_{f,x}}}{n}h^{\frac{2}{3}}\right), v_{sw,y} = \left(\frac{\sqrt{S_{f,y}}}{n}h^{\frac{2}{3}}\right) \ (5)"$$

*(2.15) Line 149-150: summarize what are those methodology and numerical methods that are used to solve the equations. You are asking the readers to read 6 other papers to get the basic information about your simulations.*

→ I believe that I am not asking the readers to read 6 other papers to get numerical methods to solve the equation described in this paper. I refer to only Kollet and Maxwell (2006). I really believe that the more details of ParFlow are not necessary for this paper. First, ParFlow has been widely adopted in the hydrologic community and very well documented previously. I have added nothing new to the ParFlow simulation in this paper. Therefore, many potential readers do not need too much details of ParFlow to understand the take-home-message of this paper. Second, I mainly focused on how data assimilation works in the hyperresolution models and I did not focus on the physics of the simulation. If I described too much details of Parflow here, the readers may be confused and cannot accurately understand the take-home-message of the paper. The original version of the paper has already been long and I did not add anything which is not absolutely necessary to understand the keypoints of the paper. I have clarified this strategy of writing at the beginning of this section. See also the response to the comment (2.16).

"Since I used this widely adopted solver as is and added nothing new to the model physics, I described the method of ParFlow to simulate integrated surface-subsurface water flows briefly and omitted the details of numerical methods."

*(2.16) Section 2. Method, line 109-150: the description of the model that is used is not informative and comprehensive. Consider restructuring and adding more details in such way that it covers the following: 1) how the unsaturated zone is modelled? 2) How surface flows are modeled? 3) How the*

*coupling of the two is being done? 4) What is the numerical approach taken for solving the equations?*

→ How the unsaturated zone is modelled has already been described in the original version of the paper.

"In the subsurface, ParFlow solves the variably saturated Richards equation in three dimensions.

$$S_S S_W(h)\frac{\partial h}{\partial t} + \phi S_W(h)\frac{\partial S_W(h)}{\partial t} = \nabla \cdot \mathbf{q} + q_r \quad (1)$$

$$\mathbf{q} = -\mathbf{K}_s(\mathbf{x})k_r(h)[\nabla(h+z)cos\theta_x + sin\theta_x] \quad (2)$$

In equation (1), $h$ is the pressure head [L]; z is the elevation with the z axis specified as upward [L]; $S_S$ is the specific storage [L$^{-1}$]; $S_W$ is the relative saturation; $\phi$ is the porosity [-]; $q_r$ is a source/sink term. Equation (2) describes the flux $\mathbf{q}$ [LT$^{-1}$] by Darcy's law, and $\mathbf{K}_s$ is the saturated hydraulic conductivity tensor [LT$^{-1}$]; $k_r$ is the relative permeability [-]; $\theta$ is the local angle of topographic slope (see Maxwell et al. 2015). In this paper, the saturated hydraulic conductivity is assumed to be isotropic and a function of z:

$$\mathbf{K}_s = K_s(z) = K_{s,surface}\exp\left(-f\left(z_{surface} - z\right)\right) \quad (3)$$

where $K_{s,surface}$ is the saturated hydraulic conductivity at the surface soil, and $z_{surface}$ is the elevation of the soil surface. The saturated hydraulic conductivity decreases exponentially as the soil depth increases (Beven 1982). A van Genuchten relationship (van Genuchten 1980) is used for the relative saturation and permeability functions.

$$S_W(h) = \frac{S_{sat} - S_{res}}{(1+(\alpha h)^n)^{\left(1-\frac{1}{n}\right)}} + S_{res} \qquad\qquad (4)$$

$$k_r(h) = \frac{\left(1 - \frac{(\alpha h)^{n-1}}{(1+(\alpha h)^n)^{\left(1-\frac{1}{n}\right)}}\right)^2}{(1+(\alpha h)^n)^{\frac{\left(1-\frac{1}{n}\right)}{2}}} \qquad\qquad (5)$$

where $\alpha$ [L-1] and n [-] are soil parameters, $S_{sat}$ is the relative saturated water content and $S_{res}$ is the relative residual saturation."

How surface flows are modelled has already been described in the original version of the paper.

"Overland flow is solved by the two-dimensional kinematic wave equation. The dynamics of the surface ponding depth, h [L], can be described by:

$$\mathbf{k} \cdot [-K_s(z)k_r(h) \cdot \nabla(h+z)] = \frac{\partial \|h,0\|}{\partial t} - \nabla \cdot \|h,0\|\mathbf{v}_{sw} + q_r \quad (4)$$

In equation (4), $\mathbf{k}$ is the unit vector in the vertical and $\|h,0\|$ indicates the greater value of the two quantities following the notation of Maxwell et al. (2015). If h < 0, equation (4) describes that vertical fluxes across the land surface is equal to the source/sink term $q_r$ (i.e., rainfall and evapotranspiration). If h > 0, the terms on the right-hand side of equation (4), which indicate

water fluxes routed according to surface topography, are active. $v_{sw}$ is the two-dimensional depth-averaged water flow velocity [LT$^{-1}$] and estimated by the Manning's law:

$$v_{sw,x} = \left(\frac{\sqrt{S_{f,x}}}{n} h^{\frac{2}{3}}\right), v_{sw,y} = \left(\frac{\sqrt{S_{f,y}}}{n} h^{\frac{2}{3}}\right) \quad (5)$$

where $S_{f,x}$ and $S_{f,y}$ are the friction slopes [-] for the x- and y-direction, respectively; n is the Manning's coefficient [TL$^{-1/3}$]. In the kinematic wave approximation, the friction slopes are set to the bed slopes."

How the coupling of the two is being done was indeed unclear in the original version of the paper. I briefly clarified this point in the revised version of the paper.

"This formulation results in the overland flow equation being represented as a boundary condition to the variably saturated Richards equation (Kollet and Maxwell 2006)."

What the numerical approach is taken for solving the equations might need long description. I omitted to explain the details and refer to literature. See the response to the comment (2.15).

*(2.17) Section 2.2. Data Assimilation, line 153-206: Please explain the topic of DA in a plain language and for audiences larger than merely DA-community. What is the general philosophy there? Why Kalman filter? Why Ensemble Transfer Kalman Filter? Is it more accurate? is it faster? Please justify the choices you made.*

→ I fully agree that I should make an effort to get audiences larger than DA-community and I thank the reviewer to help me get it done. I should admit that in the current paper, readers cannot understand what I said if they know nothing about data assimilation in advance. However, the description of data assimilation in this paper is long and detailed compared with the other related papers cited in the introduction section although I added nothing new to the existing algorithm. I believe that I provided the good guidance for potential readers to further learn data assimilation by reading the referenced review papers and textbooks.

I have added several equations to explain the problem and motivation of data assimilation. State and parameter are also defined here. The reader may understand the general philosophy here.

"The Parflow model can be formulated as a discrete state-space dynamic system:

$$x(t+1) = f\big(x(t), \theta, u(t)\big) + q(t) \quad (8)$$

where $x(t)$ is the state variables (i.e. pressure head), $\theta$ is the model parameters (i.e. saturated hydraulic conductivity), $u(t)$ is the external forcing (i.e., rainfall and evapotranspiration), and

$q(t)$ is the noise process which represents the model error. In data assimilation, it is useful to formulate an observation process as follows:

$$y^f(t) = \mathcal{H}\big(x(t)\big) + r(t) \qquad\qquad (9)$$

where $y^f(t)$ is the simulated observation, $\mathcal{H}$ is the observation operator which maps the model's state variables into the observable variables, and $r(t)$ is the noise process which represents the observation error. The purpose of EnKF (and any other data assimilation methods) is to find the optimal state variables $x(t)$ based on the simulation $y^f(t)$ and observation (defined as $y^o$) considering their errors ($q(t)$ and $r(t)$)"

I chose (ensemble) Kalman filter since it is widely used in the community. I have already discussed this point in the response to the comment (2.2). See the response to the comment (2.2). I show the revised sentence in the response to the comment (2.2) again below.

"The EnKF has widely been applied to hyper-resolution land models (e.g., Camporese et al. (2009); Camporese et al. (2010); Ridler et al. (2014); Zhang et al. (2015); Kurtz et al. (2016); Zhang et al. (2018)). I examine if surface lateral flows matter for data assimilation of soil moisture observations into hyperresolution land models using this widely adopted data assimilation method. "

ETKF is in the family of EnKF. The original EnKF tells us only mean and variance of the targeted variables and do not tell how to modify each individual ensemble. ETKF is an algorithm to tell us how to transfer each ensemble. Therefore, ETKF is necessary to continue the cycles of data assimilation. This point has already included in the original version of the paper.

"Although equations (6-11) can calculate the mean and variance of the ensemble members, they do not tell how to adjust the state of the ensemble members in order to realize the estimated mean and variance. There are many proposed flavors of EnKF and one of the differences among them is the method to choose the analysis $x_i^a$. In this paper, the Ensemble Transform Kalman Filter (ETKF; Bishop et al. 2001; Hunt et al. 2007) was used to transport forecast ensembles to analysis ensembles. Please refer to Hunt et al. (2007) for the complete description of the ETKF and its localized version, the Local Ensemble Transform Kalman Filter (LETKF)."

In the revised version of the paper, I mentioned that ETKF is used in the previous hyperresolution land data assimilation.

"ETKF has been used for hyperresolution land data assimilation (e.g., Kurtz et al. 2016)."

*(2.18) Line 193-194: what do you mean? Please explain what an "ensemble inflation method" is.*

*Explain how this method correct for this?*

→ In many EnKF systems, the ensemble spread, $P^a$, will be too small to stably implement data assimilation cycles due to a non-linear function of the background error in Kalman filter (why the ensemble spread becomes too small was omitted for brevity). The ensemble inflation method arbitrary inflates the ensemble spread to maintain the filter. This point was indeed unclear in the original version of the paper. I have clarified this point in the revised version of the paper.

> "In many ensemble Kalman filter systems, the ensemble spread, $P^a$, tends to become too underdispersive to stably perform data assimilation cycles without any ensemble inflation methods (Houtekamer and Zhang, 2016). To overcome this limitation, $P^a$ is arbitrarily inflated after data assimilation."

*(2.19) Line 195: again you are throwing a term and you are asking your audience to read an entire paper to understand what "relaxation to prior perturbation method (RTPP)" is!*

→ I am not asking to read Zhang et al. 2004 because RTPP is equation (12). The reviewer speculated that RTPP is a very complicated scheme, but I believe that no further explanation is required. I have decided not to change this aspect of the paper.

*(2.20) Line 199: where does this number come from?        Report the literature or justify your choice.*

→ If alpha is set to 1, I did not decrease the spread by data assimilation. Many previous studies set alpha close to 1 (i.e. ensembles are strongly inflated) so that I also used a large alpha. This point was indeed unclear in the original version of the paper. I have clarified this point in the revised version of the paper.

> "If $\alpha = 1$, the analysis spread is identical to the background spread. Many studies show that the ensemble inflation works well when $\alpha$ remains fairly close to 1 (see also the comprehensive review by Houtekamer and Zhang 2016)."

*(2.21) Line 201-206: Badly written. It is not clear which observations are assimilated in the system. It is not clear what are the state variables that are updated in every modeling time step.*

→ Pressure head is updated in the data assimilation system. It was clarified in the original version of the paper:

> "In the data assimilation experiments, I adjusted pressure head by data assimilation so that x^f is pressure head."

The other reviewer also pointed out that this description was not comprehensive. I clarified that what the observation is in the revised version of the paper:

> "I assimilated volumetric soil moisture observations so that $y^f$ and $y^o$ are simulated and observed volumetric soil moisture, respectively. The van Genuchten relationship converts the adjusted state variables $x^f$ to the observable variables $y^f$ and can be recognized as an observation operator $\mathcal{H}$. However, since volumetric soil moisture $y^f$ has already been calculated by Parflow, I did not need the van Genuchten relationship in data assimilation."

*(2.22) Line 210-217: Please better explain what kind of measure "KLD" is, what it measures and why you used this measure to quantify the non-gaussianity of the background error? What do you do with the asymmetry of KLD? Is P Gaussian distribution or q?*

→ KLD measures the closeness of two probabilistic distribution functions (PDF). This point was mentioned in the original version of the paper:

> "A large value for $D_{KL}(p, q)$ indicates that the two PDFs, $p$ and $q$, substantially differ from each other. Therefore, the KLD can be used as an index to evaluate the closeness of two PDFs."

The KLD has been used to evaluate the Gaussianity of the sampled background error in the context of data assimilation. This point was indeed unclear in the original version of the paper. I have clarified this point in the revised version of the paper:

> "The KLD has been used to quantitatively evaluate the Gaussianity of the sampled background error in the studies on data assimilation (e.g., Kondo and Miyoshi 2019; Duc and Saito 2018)."

The asymmetry of KLD does not matter in this study since I always set p to the PDF of ensembles and set q to the theoretical Gaussian PDF which has the mean and variance of the targeted ensembles. It also answers the final question of the reviewer. In the original version of the paper, this point has been mentioned in the results section (not this section):

> "I calculated KLD by comparing the PDF of the NoDA ensemble ($p$ in equation (13)) with the Gaussian PDF which has the mean and variance of the NoDA ensemble ($q$ in equation (13))."

I realized that this is confusing. In the revised version of the paper, I have clarified this point in the section 2.3:

> "In this study, I compared the PDF of the ensemble simulation (p in equation (13)) with the Gaussian PDF which has the mean and variance of the ensembles (q in equation (13)). A large value for $D_{KL}(p, q)$ indicates the state variables simulated by ensembles do not follow the

Gaussian PDF."

(2.23) *Line 235: where did you get these estimates? Provide references or justify your choice.*

→ Since this is not the real-data experiment, I believe that I need not to justify the parameter variables, which were basically arbitrarily chosen. I have clarified that our specified parameters were in the reasonable range.

> "Those values are in the reasonable range estimated by the published literature (e.g., Ghanbarian-Alavijeh et al. 2010)."

(2.24) *Section 3.1.1 experimental design: what do you want to achieve with these two synthetic experimental design? Please explain why you made your choices of experimental design more clearly. e.g., Why dividing your domain to rain and no-rain regions?*

→ The purpose of this experiment is to show how surface lateral flows matter in hyper-resolution hydrologic data assimilation as I described in the original version of the paper:

> "The synthetic experiment was implemented to examine how topography-driven surface lateral flows contribute to efficiently propagating observation's information horizontally in the data assimilation of soil moisture observation."

To achieve this goal, I set up a simplified experiment eliminating any uncertainties which are not related to our purpose. I divided the domain into rain and no-rain regions because the effect of surface lateral flows generated from the upper rain area on data assimilation can clearly be discussed. Although this setting might be oversimplified and unrealistic, I did another experiment in which the distribution of rainfall is more realistic (section 3.2). This point was indeed unclear in the original version of the paper and I have clarified this point in the revised version of the paper:

> "Although this rainfall distribution is unrealistic, the effect of surface lateral flows on data assimilation can clearly be discussed in this simplified problem setting. More realistic rainfall distribution will be used in the next synthetic experiment (see section 3.2)."

(2.25) *Line 268-269: most land surface models use satellite observations of surface soil moisture for DA. Observations are usually not available to the depth of 1m! Please explain more clearly what you want to gain by assimilating SM at several depth?*

*How reference runs are different from no-assimilation runs?*

→ In in-situ observations, deeper-zone soil moisture is often observed. In the spatial scale focused in

this paper, those in-situ observations might be useful. This point was indeed unclear in the original version of the paper, and I have mentioned this point in the revised version of the paper:

> "These soil moisture observations can be obtained in the in-situ observation sites (e.g., Dorigo et al., 2016)."

I fully agree with the reviewer's comment that most land surface models use satellite observation of surface soil moisture so that it is reasonable to assume only surface soil moisture can be observed. Please note that I used only surface soil moisture observations in the section 3.2, which might be more realistic:

> "Unlike the experiment in section 3.1, only surface soil moisture can be observed in this synthetic experiment, which makes this experiment more realistic since satellite sensors can observe only surface soil moisture."

In the revised version of the paper, I mentioned this point in the first experiment (section 3.1) in order to tell the readers that I will do the more realistic experiment.

> "In the section 3.2, I will assume that only surface soil moisture observation can be accessed, which is more realistic since satellite sensors can observe only surface soil moisture."

The reference run is a single model run with the specified single rainfall rate and saturated hydraulic conductivity. On the other hand, in the NoDA experiment, it is assumed that I did not know this specified rainfall and conductivity (synthetic true). The no-assimilation runs are ensemble model runs with uncertainty in rainfall and saturated hydraulic conductivity. This point was indeed unclear in the original version of the paper and I have clarified this point in the revised version of the paper:

> "Please note that in the NoDA experiment, the true rainfall rate and saturated hydraulic conductivity were unknown so that I could not accurately estimate the synthetic true state variables. I will evaluate how this negative impact of uncertainties in rainfall and saturated hydraulic conductivity can be mitigated by data assimilation in the DA experiment."

*(2.26) Line 322: What do you mean by Ks,surface is accurate?*

→ Since our synthetic truth of Ks was set to 0.005 [m/h] and our estimated Ks by data assimilation was 0.00508, our estimation was accurate. I rephase this sentence in order to accurately deliver this point to the readers:

> "The estimated $K_{s,surface} \approx 0.00508$ [m/h] by ETKF is mostly identical to the synthetic truth."

> "The estimated $K_{s,surface} \approx 0.00512$ [m/h] by ETKF is mostly identical to the synthetic truth."

> "The saturated hydraulic conductivity estimated by ETKF is mostly identical to the synthetic

truth ($K_{s,surface} \approx 0.0204$ [m/h]).”

*(2.27) Line 322: the increase of RSME or IR index?*

→ It should be the increase of RMSE. I have decided not to change this aspect of the paper.

*(2.28) Line 324-326: check the structure of the sentence.*

→ I have modified this sentence as follows:

> “The impact of data assimilation on the surface soil moisture simulation is small because the volumetric soil moisture's RMSE of the NoDA experiment in this surface soil layer is already small ($\leq 0.01 \text{m}^3/\text{m}^3$) in the case of the LOW_K reference so that any improvements do not make sense.”

*(2.29) Line 325: RMSE of what variable?*

→ RMSE of volumetric soil moisture. This definition can be found in the original version of the paper:

> “As evaluation metrics, root-mean-square-error (RMSE) was used:
>
> $$\text{RMSE} = \sqrt{\frac{1}{k}\sum_{i=1}^{k}(F_i - T)^2} \quad (14)$$
>
> where k is the ensemble number, $F_i$ is the volumetric soil moisture simulated by the i-th member in the DA or NoDA experiment, T is the volumetric soil moisture simulated by the synthetic reference run.”

I have clarified this point once again here in the revised version of the paper:

> “The impact of data assimilation on the surface soil moisture simulation is small because the volumetric soil moisture's RMSE of the NoDA experiment in this surface soil layer is already small ($\leq 0.01 \text{m}^3/\text{m}^3$) in the case of the LOW_K reference so that any improvements do not make sense.”

*(2.30) Line 329-330: this statement is not correct!*

→ I guess that the reviewer is worrying about the difference at the left edge of the domain. I have noticed and mentioned it in the original version of the paper:

> “Although observing the lower part of the slope slightly improves the soil moisture simulation at the left edge of the domain compared with observing the upper part of the slope (the reason for it

will be explained later), there are few differences between the UP_O and DOWN_O scenarios in the case of the LOW_K reference"

This difference at the left edge of the domain can be explained by non-linear and non-Gaussian processes as the skill's degradation at the other places. This point has also been mentioned in the original version of the paper:

> "Please note that the non-Gaussianity can also be found in the LOW_K reference at the edge of the domain (x=500m) due to the non-linear dynamics, which causes the degradation of the soil moisture simulation in the LOW_K-UP_O experiment (see Figure 2a)."

I noticed that this point should also be mentioned in the lines 329-330. I have modified this point in the revised version of the paper.

> "The IR's spatial pattern of the LOW_K-DOWN_O experiment is similar to that of the LOW_K-UP_O experiment except for the left edge of the domain."

*(2.31) Line 332: where does this optimized value for K_sat come from? How do you infer its accuracy?*
→ Ks was estimated by ETKF. I noticed that the phrase "optimized" may be misleading. In the revised version of the paper, I explicitly mentioned that our Ks was the estimated values by ETKF. See also the response to the comment (2.26).

> "The estimated $K_{s,surface} \approx 0.00508$ [m/h] by ETKF is mostly identical to the synthetic truth."
>
> "The estimated $K_{s,surface} \approx 0.00512$ [m/h] by ETKF is mostly identical to the synthetic truth."
>
> "The saturated hydraulic conductivity estimated by ETKF is mostly identical to the synthetic truth ($K_{s,surface} \approx 0.0204$ [m/h])."
>
> "Although the observations in the lower part of the slope (see the black arrow in Figure 2d) significantly contribute to improving the soil moisture simulation in the downstream area of the observation and accurately estimating $K_{s,surface} \approx 0.0208$ [m/h], the impact of the data assimilation on the shallow soil moisture simulation around x=500~1000m is marginal."

*(2.32) Line 335-336: Please elaborate what this difference indicates.*
→ What this difference indicates can be found later in the original version of the paper:

> "Please note that the non-Gaussianity can also be found in the LOW_K reference at the edge of the domain (x=500m) due to the non-linear dynamics, which causes the degradation of the soil moisture simulation in the LOW_K-UP_O experiment (see Figure 2a)."

Please also see the response to the comment (2.30). Every difference of IR can be explained by the same mechanism. Since the difference shown by Figure 3b is more significant than that by Figure 3a, I emphasized more on the difference shown by Figure 3b and explained the difference appears more often in the case of higher hydraulic conductivity. In the revised version of the paper, I have clarified that what this difference indicates will be explained later:

> "Although observing the lower part of the slope slightly improves the soil moisture simulation at the left edge of the domain compared with observing the upper part of the slope (the reason for it will be explained later), there are few differences between the UP_O and DOWN_O scenarios in the case of the LOW_K reference."

*(2.33) Line 342: How do you explain figure 2a then? The valley bottom in figure 2a (left corner) should theoretically improve when lateral transfers are included (when the uphill is raining). This figure shows that the observations of SM (also in depth) didn't improve the predictions downhill although lateral transfer processes are included in your model!*

→ I guess that the reviewer thought that I switched on and off the lateral transfer processes in this experiment. I did not do it here. I compared the DA and NoDA experiment. Both of them used the full ParFlow which included lateral transfer processes. I just switched on and off data assimilation in this experiment. Although the model has lateral transfer processes, the NoDA experiment does not have a good skill since I do not know the true rainfall rate and saturated hydraulic conductivity in the NoDA experiment (see Figure S3, for instance). Despite uncertainty in rainfall and saturated hydraulic conductivity, I can improve the simulation skill almost everywhere except for the left edge of the domain by assimilating volumetric soil moisture at the single location. This is what Figure 2a tells us. First, I have clarified that the NoDA experiment do have a lot of uncertainty and mitigating this uncertainty is the purpose of the DA experiment in the revised version of the paper: See also the response to the comment (2.25).

> "Please note that in the NoDA experiment, the true rainfall rate and saturated hydraulic conductivity were unknown so that I could not accurately estimate the synthetic true state variables. I will evaluate how this negative impact of uncertainties in rainfall and saturated hydraulic conductivity can be mitigated by data assimilation in the DA experiment."

Second, I have repeated the sources of error when I explained Figure 2a in the revised version of the paper:

> "Despite the uncertainty in rainfall and hydraulic conductivity, RMSE is reduced by data assimilation not only directly under the observation but also the lower part of the slope where it does not rain."

*(2.34) Line 356-363: I am confused! How did you conclude this? Didn't you assimilate SM observations across all areas (where it rains and where it does not rain)? The DA results are not sufficiently and clearly explained.*

→ No, I did not assimilate SM observation across all areas. I believe that this point was written in the original version of the paper (see also Table 1).

> "The two scenarios of the observation's location are provided. In the first scenario (hereafter called the UP_O scenario), the volumetric soil moisture at the upper part of the slope (x = 2500m) was observed. In the UP_O scenario, I could observe the volumetric soil moisture in the upper part of the slope where it heavily rains and tried to infer the soil moisture in the lower part of the slope where it does not rain by propagating the observation's information downhill. In the second scenario (hereafter called the DOWN_O scenario), the volumetric soil moisture at the lower part of the slope (x = 1500m) was observed. In the DOWN_O scenario, I could observe the volumetric soil moisture in the lower part of the slope where it does not rain and tried to infer the soil moisture in the upper part of the slope where it heavily rains by propagating the observation's information uphill."

In the revised version of the paper, I emphasized that I did not assimilate SM observation across all areas.

> "I assumed that the small part of the domain can be observed."

*(2.35) Line 365-366: what do you mean?*

→ I meant that the small difference of RMSE between two observation scenarios shown in Figure 3a cannot be found in Figure 3b. I have clarified this point in the revised version of the paper.

> "The high representativeness of the observations which I found in the case of the LOW_K reference (i.e. the small difference of RMSEs between two observation scenarios) cannot be found in the case of the HIGH_K reference."

*(2.36) Line 370-371: yes, because of higher Hydraulic conductivity!*

→ I guess that the reviewer agreed with this point. I have decided not to change this aspect of the paper.

*(2.37) Line 395: nonlinear dynamics of what?*

→ nonlinear dynamics of surface lateral flows. I have clarified this point in the revised version of the paper.

> "Please note that the non-Gaussianity can also be found in the LOW_K reference at the edge of the domain (x=500m) due to the non-linear dynamics of surface lateral flows, which causes the degradation of the soil moisture simulation in the LOW_K-UP_O experiment (see Figure 2a)."

*(2.38) Line 409: you don't show that the errors are correlated.*

→ This statement was indeed misleading. I would like to say that the state adjustment (using background error correlation in data assimilation) has a significant role in improving the soil moisture simulation compared with the parameter estimation. I have clarified this point in the revised version of the paper:

> "The horizontal propagation of the observations' information shown in this experiment was brought out not only by the estimation of spatially homogeneous saturated hydraulic conductivity but also by the adjustment of state variables (i.e., pressure head and volumetric soil moisture)."

*(2.39) Line 412-413: This is what most land surface models do when they assimilate SM and they still show improvements!*

→ Yes, they show improvement. This is because their spatial resolution was often larger than 25km. In this spatial scale, surface lateral flows (or any lateral flows) are not visible. They showed improvement in this coarse spatial scale, but they could tell nothing about the performance to simulate soil moisture in the finer scales which explicitly resolve topography-driven surface lateral flows. I did not focus on these conventional land surface models. Here I focus on the data assimilation methodology for the next generation hyperresolution land surface model. I believe that this motivation has been clarified in the introduction section (and title) of the original paper:

> "While conventional land surface models (LSMs) assume that lateral water flows are negligible at the coarse resolution (>25km) and solve vertical one-dimensional Richards equation for the soil moisture simulation (e.g., Sellers et al. 1996; Lawrence et al. 2011), currently proposed hyperresolution land models, which can be applied at a finer resolution (<1km), explicitly consider surface and subsurface lateral water flows (e.g., Maxwell and Miller 2005; Tian et al. 2012; Shrestha et al. 2014; Niu et al. 2014)."

> "The fine horizontal resolution can resolve slopes, which are drivers of a lateral transport of water, and realize the fully integrated surface-groundwater modeling."

See also the response to the comment (2.6). This point was also mentioned in the discussion section:

> "This potential cannot be brought out in the conventional 1-D LSM where sub-grid scale surface runoff is parameterized and the surface flows in one grid do not move to the adjacent grids."

As the response to this comment, I briefly explain this point again in the lines 412-413:

> "Please note that although many conventional land surface models neglected or parameterized lateral flows, this assumption can be applied only in the coarse spatial resolution (>25km), which is not the case of this experimental setting."

*(2.40) Line 435: why is this constant number added? Please justify the choices you made.*

→ Variogram gives us the covariance field. I added the constant number to regulate the mean of logarithm of saturated hydraulic conductivity. This procedure has been done in the previous studies (e.g., Kurtz et al. 2016). This point was indeed unclear in the original version of the paper. I have clarified this point in the revised version of the paper:

> "A constant value of -2.30 $log_{10}$(m/h) (i.e. 0.005 (m/h)) was added to the generated field so that the mean of the logarithm of surface saturated hydraulic conductivity was set to -2.30 (i.e. 0.005(m/h)). This method to generate the field of the saturated hydraulic conductivity has been used previously (e.g., Kurtz et al. 2016)."

*(2.41) Line 467: what is "OF"?*

→ Overland Flow. I have clarified this point in the revised version of the paper:

> "In the first configuration, called OF (Overland Flow), Parflow explicitly solves overland flows."

*(2.42) Line 488-492: weak justification. What is over fitting? what is wrong with that? why does it have a negative impact? elaborate.. also in lines 505-507.*

→ In the revised version of the paper, I rephased this point for those who are not familiar to this technical term in machine learning.

> "This is because the adjustment of hydraulic conductivity in the OF_DA_obs361 experiment greatly mitigates not only the errors induced by uncertainty in hydraulic conductivity but those induced by uncertainty in rainfall rate. In the OF configuration, there are two sources of errors, rainfall rate and hydraulic conductivity. However, data assimilation can adjust only hydraulic conductivity in this study. Although it is expected that the adjustment of hydraulic conductivity

mainly mitigates the errors of simulated volumetric soil moisture induced by uncertainty in hydraulic conductivity, it also greatly mitigates those induced by uncertainty in rainfall rate by adjusting the parameter in the incorrect direction when the number of observations is large. Therefore, the assimilation of a large number of observations degrades the estimation of saturated hydraulic conductivity despite the improvement of the soil moisture simulation."

"The assimilation of a large number of observations degrades the estimation of saturated hydraulic conductivity because it greatly mitigates the impact of all systematic errors which comes from three different sources only by adjusting hydraulic conductivity."

*(2.43) Line 522-525: what variable are you talking about?*

→ Ensemble simulation of volumetric soil moisture. I have clarified this point in the revised version of the paper:

"Figure 10a clearly shows that the ensemble simulation of volumetric soil moisture generates the strong non-Gaussianity at the edge of the area where topography-driven surface flow reaches, which harms the efficiency of the ETKF. This finding is consistent to what I found in the previous experiment in section 3.1."

*(2.44) Line 533-535: then this assimilation technique is not suitable for topographically driven systems.*

→ Please note that in most areas of my experiment, ETKF efficiently reduced RMSE in the two synthetic experiments as it does for the 1-D conventional land surface model with a coarse resolution. However, I found the limitation of ETKF which cannot be found if you work only with the conventional 1-D land surface model. I believe that it is too aggressive to say ETKF is not suitable since it can reduce RMSE in most cases and many previous studies found it is useful. In the revised version of the paper, I clarified that ETKF is effective, but have some limitations which cannot be found in the conventional 1-D land modeling:

"Although ETKF can significantly improve the simulation skill of the hyperresolution land model in many cases, I found that it has limitations when it is applied to the problems with the topography-driven surface lateral flows. Figure 10 clearly indicates that this limitation appears only if lateral water flows are explicitly considered."

See also the response to the comment (2.2). Although the reviewer may think that the limitation of ETKF I found is the fatal flaw of this paper, this finding is actually one of the major achievements of this study. I do not propose the new data assimilation method. ETKF (and EnKF) has been widely

used in the communities of both the conventional 1-D LSM and the hyperresolution 3-D models. In this paper, I discussed how this widely adopted EnKF works with surface lateral flows, which has been overlooked in the previous studies, and found this limitation. I made some modifications to clarify this point as the response to the comment (2.2).

*(2.45) Line 553, add a reference to this argument.*

→ This is the results of this study. I clarified this point in the revised version of the paper:

> "==I found that== neglecting the topography-driven surface flow causes significant bias in the soil moisture simulation and this bias cannot be completely mitigated by data assimilation especially in the case of a sparse observing network."

*(2.46) Line 564-566: knowing that Gaussian distribution of background error is a pre-requisite, why this method then?*

→ This comment is strongly related to the comments (2.2) and (2.44). See the responses to the comments (2.2) and (2.44). I am not the first person who chose this method. Many previous works chose this method neglecting the matter which I pointed out in this paper. I highlighted the importance of surface lateral flows in hydrological data assimilation and provided the future direction of the algorithm development. Although this point was clarified by the responses to the comment (2.2) and (2.44), I further added a sentence to clarify this point in line 564-566 of the revised version of the paper.

> "The conventional ensemble data assimilation (i.e. ETKF) severely suffers from the non-Gaussian background error PDFs caused by the strongly nonlinear dynamics of the topography-driven surface flow ==although it has been widely used by previous studies (e.g., Camporese et al. (2009); Camporese et al. (2010); Ridler et al. (2014); Zhang et al. (2015); Kurtz et al. (2016); Zhang et al. (2018))==."

*(2.47) Line 594: what is localization method?*

→ The localization method is the group of methodology to restrict the impact of assimilating observations. In this paper, assimilating observation impacted everywhere in the computational domain. If the localization method is applied, assimilating an observation influences state variable of the model grids which are near to the location of observations. This point was indeed unclear in the original version of the paper and I have clarified this point in the revised version of the paper:

"In this study, assimilating observation impacted everywhere in the computational domain. If the localization method is applied, assimilating observation influences state variables of the model grids which are near to the location of assimilated observations."

*(2.48) Bottom line: I don't understand the choice of experimental set up, the data assimilation technique chosen given the non-gaussianity of the background error, and the justification of the simulation results.*

→ The choice of experimental setup: see the responses to the comments (2.5), (2.6), (2.7), (2.12), (2.24) and (2.25).

The data assimilation technique chosen given the non-gaussianity of the background error: see the responses to the comments (2.2), (2.5), (2.10), (2.17), (2.18), (2.19), (2.20), (2.21), (2.44), (2.46), and (2.47).

The justification of the simulation results: see the responses to the comments (2.26), (2.27), (2.29), (2.30), (2.31), (2.33), (2.34), (2.35), (2.36), (2.37), (2.38), (2.39), and (2.42).

*Minor comments:*

*(2.49) Consider breaking long sentences into multiple sentences. Examples of this include: Line 26-30, 55-61, 69-74*

→ I have modified some sentences following the reviewer's instruction. For instance,

"It is difficult to efficiently constrain model states at the edge of the area where the topography-driven surface flow reaches by linear-Gaussian filters. It brings the new challenge in land data assimilation for hyperresolution land models."

*(2.50) Line 111, what do you mean by "parallel simulation platform"?*

→ I have rephrased it as follows:

"This model can be efficiently parallelized in high performance computers and has been widely used as a core hydrological module in hyperresolution land models (e.g., Maxwell and Kollet 2008; Maxwell and Condon 2016; Fang et al. 2017; Kurtz et al. 2016; Maxwell et al. 2011; Williams and Maxwell 2011; Shrestha et al. 2014)."

*(2.51) Line 113, provide examples of such land models.*

→ I believe that the examples are given in the parenthesis:

"(e.g., Maxwell and Kollet 2008; Maxwell and Condon 2016; Fang et al. 2017; Kurtz et al. 2016; Maxwell et al. 2011; Williams and Maxwell 2011; Shrestha et al. 2014)"

*(2.52) Line 160-171: what is Pa? In general, what do "a" and "f" stand for?*
→ This point was indeed unclear in the original version of the paper. I have clarified this point in the revised version of the paper:

"Superscripts f and a are forecast and analysis, respectively."

"$P^a$ is an updated analysis error covariance."

*(2.53) Line 647: space after error*
→ I have deleted it.